# Remote Dyslexia Screening for Bilingual Children

Maren Eikerling [1,2], Matteo Secco [3], Gloria Marchesi [4], Maria Teresa Guasti [2], Francesco Vona [3], Franca Garzotto [3] and Maria Luisa Lorusso [1,*]

1. Unit of Neuropsychology of Developmental Disorders, Scientific Institute IRCCS E. Medea, 23842 Bosisio Parini, Italy; m.eikerling@campus.unimib.it
2. Department of Psychology, Piazza Ateneo Nuovo 1, University Milan-Bicocca, 20126 Milan, Italy; mariateresa.guasti@unimib.it
3. Department of Electronics, Information and Bioengineering, Politecnico di Milano, Via Ponzio 34/5, 20133 Milan, Italy; matteo.secco@mail.polimi.it (M.S.); francesco.vona@polimi.it (F.V.); franca.garzotto@polimi.it (F.G.)
4. Department of Brain and Behavioural Sciences, University of Pavia, Via Agostino Bassi, 21, 27100 Pavia, Italy; gloriamarchesi3@gmail.com
* Correspondence: marialuisa.lorusso@lanostrafamiglia.it

**Abstract:** Ideally, language and reading skills in bilingual children are assessed in both languages spoken in order to avoid misdiagnoses of communication or learning disorders. Due to limited capacity of clinical and educational staff, computerized screenings that allow for automatic evaluation of the children's performance on reading tasks (accuracy and speed) might pose a useful alternative in clinical and school settings. In this study, a novel web-based screening platform for language and reading assessment is presented. This tool has been preliminarily validated with monolingual Italian, Mandarin–Italian and English–Italian speaking primary school children living and schooled in Italy. Their performances in the screening tasks in Italian and—if bilingual—in their native language were compared to the results of standardized/conventional reading assessment tests as well as parental and teacher questionnaires. Correlations revealed the tasks that best contributed to the identification of risk for the presence of reading disorders and showed the general feasibility and usefulness of the computerized screening. In a further step, both screening administrators (Examiners) and child participants (Examinees) were invited to participate in usability studies, which revealed general satisfaction and provided suggestions for further improvement of the screening platform. Based on these findings, the potential of the novel web-based screening platform is discussed.

**Keywords:** bilingualism; developmental dyslexia; child reading acquisition; computerized screening; usability; web application; remote testing

## 1. Introduction

### 1.1. Developmental Dyslexia in Multilingual Children

Multilingual persons are confronted with more than one language in their everyday life [1]. Since the language used in the family context often is the first language acquired, it is referred to as L1, while the language used outside the home is referred to as the societal language or L2. In multilingual families and in contexts where language mixing and switching is the rule, the different languages can also be easily acquired simultaneously [2]. Language competence can vary between the languages spoken.

Generally speaking, multilingual language acquisition does follow the same stages as monolingual language acquisition but can vary in terms of timing [3]. Due to the high variability in language and orthographic systems and to the bidirectional influences occurring among them, literacy acquisition in bilingual contexts is characterized by great heterogeneity and this may pose challenges for the identification of specific language and reading disorders [4].

Since reading is a complex, multifaceted construct, comprising a number of subskills, the acquisition of which necessitates distinct linguistic knowledge, learning to read in the L2 implies subskills development that involves two languages and the metalinguistic knowledge that must refer to two languages [5]. Indeed, depending on the similarities and differences between the languages, bilingualism may induce facilitation or interference processes [6].

In fact, even when using only one of their languages, bilinguals often access linguistic and orthographic representations of their other language [7]. It has been well-established that the interaction between the two languages can induce bidirectional "transfer": the competence gained in one language, especially in sequential acquisition, is used to facilitate reading acquisition in the other language [8,9].

Formal learning to read and write usually starts at five to six years of age, although the cognitive bases for reading acquisition are built long before school start [10]. Some children however struggle in the acquisition and development of reading and writing skills. These difficulties may be due to "Specific Reading Disorders" or "Developmental Dyslexia" (DD,) affecting about 3% to over 10% of school-aged children (depending on the degree of orthographic transparency of the language) [11,12].

Affected children's problems are manifested by both reading errors and a slowed reading pace [13]. The ability to transfer auditory information into written (or vice versa) is almost always affected in dyslexic children [14] and phonological skills are regarded as predictors of dyslexia in mono-and multilingual dyslexic children [15], even if their relative weight may vary between different language systems [9].

The identification of DD is not trivial, especially in the context of multilingual children. While a series of standardized tests are applied for the diagnosis of monolingual children, assessing the reading skills in multilingual children can be challenging due to heterogeneous contexts of language acquisition (amount and quality of exposure)—diagnostic material normed on monolingual children may lead to multilingual children being incorrectly identified as having DD [16]. Indeed, reading and writing acquisition normally occurs in the societal language (the L2) and at school age, when also early bilinguals have usually reached a good mastery of their L2.

Nonetheless, the mapping of sounds to symbols, which constitutes the essence of learning to read, is necessarily influenced by other linguistic abilities, such as phonological and metaphonological skills. Moreover, reading (meaningful text) is facilitated by lexical, semantic and morphosyntactic abilities. All these language skills will reflect the impact of bilingualism [17], and assessment needs to take into account the possible effects of cross-linguistic interference. De Lamo White & Jin [18] point to biases in language assessment that penalize multilingual children.

Several different approaches aim at the reduction of such biases: (1) contextualization of test results based on length of exposure and age of onset of the L2 [19,20], (2) availability of standardized testing material that provides norms for multilingual children [21] or (3) testing reading in both languages spoken by the child [22]. Since text comprehension, but not the acquisition of decoding skills in single words or simple sentences requires high proficiency in the L2, it is recommended that a multilingual child without cognitive impairment showing persistent difficulties in decoding may undergo a diagnostic process in which also the child's L1 is taken into consideration [23,24].

Use of computerized, gamified (i.e., where the tasks are made as similar as possible to a game) testing material provides an opportunity to easily implement all these strategies and further offers the possibility to (a) enhance children's motivation [25] and (b) operate independently of pen-and-paper or flashcard testing material and remotely [26]. Previous studies [27,28] showed that the results of computerized reading and reading-related tasks are significantly associated with reading performance measured with standardized reading tests. In order to support diagnostic decisions, so-called "clinical markers" of DD can be very helpful.

DD diagnosis is based on standardized reading tests, which typically assess a child's reading accuracy and speed [29,30]. In addition to that, correlations between reading abilities and performance in further linguistic domains have been found. Although the presence of further weaknesses apart from reading- and writing-related ones does not belong to diagnostic criteria, it can provide useful information with respect to the presence and nature of specific reading disorders (thus acting as clinical markers) and indicate further needs and rehabilitation goals for a specific child.

Typically, dyslexic children show deficits in phonological awareness [31], i.e., the capacity of manipulating and analyzing the sounds that constitute words, regardless of meaning [32]. Da Silva et al. [33] confirmed these findings independently of the language and orthographic system a child acquires.

Studies across different languages have further shown that children with or at-risk for DD very often show poorer performance in Rapid Automatized Naming (RAN) compared to typically developing peers [34,35]. In such tasks, children are requested to name a limited set of visually displayed familiar objects, colors or digits, as fast as they can, throughout repeated presentations. Longitudinal studies showed that speed in RAN tasks is associated with later reading abilities [36–38].

Associations between children's reading abilities and their grammatical abilities (morphology and morphosyntax) have been found in some studies [39,40]. In Italian, difficulties in subject–verb agreement [41] and in the production or comprehension of clitic pronouns [42,43] have been described as language-related clinical markers. In English, past tense is considered a clinical marker for Developmental Language Disorder (DLD), which in turn is often associated with DD [44,45]. Past tense use discriminates bilingual children from children with DLD [46,47].

### 1.2. The MuLiMi Screening Platform

MuLiMi is a web platform developed by the Scientific Institute "E. Medea" in collaboration with Politecnico Milano with the aim to provide a computerized battery of screening tests for language and reading disorders in multilingual children. It provides three different subsystems, each one addressing a specific category of users: (1) Administrators, in charge of configuring and updating tests; (2) Examiners, who supervise test execution and provide an interpretation of the results; and (3) Examinees, who are typically young children who will undergo testing through the platform.

Each subsystem is characterized by different interfaces. The Administrator interface aims to minimize the time and manual work needed to create and customize new tests; the Examiner interface attempts to minimize the learning curve required to execute and evaluate tests; and the Examinee interface is designed to make the whole testing procedure as enjoyable as possible for the Examinee (see Section 2.2 for a detailed description).

In an attempt to meet research needs in spite of the restrictions imposed by the COVID-19 pandemic, the platform was adapted to also allow execution of the tests at a distance. This remote testing feature has proven to be very valuable even after the peak of the pandemic, removing the need for the Examinee to move to the Examiner's location for testing. This possibility encouraged participation in research, reducing organizational burdens, such as the costs and time needed for transfer. Remote testing is made possible in MuLiMi by exploiting the WebRTC technology, an open-source tool developed by Google to facilitate the development of peer-to-peer web applications.

### 1.3. Research Goals and Hypotheses

The present study describes the possible contribution of MuLiMi to DD risk identification in bilingual children. The main aim of the study was to investigate the applicability and user-friendliness of fully computerized, web-based bilingual reading screenings. Furthermore, it aimed to provide a preliminary analysis of clinical applications for dyslexia risk identification in bilingual children attending primary schools in Italy.

Precisely, in order to assess concurrent validity, standardized tests measuring similar skills were administered, and their results were compared to the screening tasks' results. To this aim, based on the standardized tests, a risk score was generated to characterize the status of each child in terms of varying degrees of risk to suffer from a specific learning disorder.

Such a risk score had no diagnostic purpose but was created to assess the screening's ability to differentiate different levels of risk in the population. Furthermore, the levels of risk for DD as judged by the children's parents and/or teachers and expressed on ad-hoc questionnaires were compared with the screening outcomes. Finally, in order to validate not only the content but also the format of the screening and its administration, usability studies were carried out with children who were administered the screening remotely and the Examiners who in turn administered it.

It was expected that:

- Children's performance in standardized/traditional reading tests would be associated with their performance in computerized screening tasks that are declared to be measuring the same construct (concurrent validity).
- Children at-risk of DD (risk score based on their performance in traditional/standardized reading tests and the parents' and teachers' evaluation of risk factors) would perform worse in the L1 and L2 computerized screening tasks than typically developing children (discriminant validity).
- The newly developed platform would respond to the needs and expectations of users (Examinees/child participants and Examiners).

## 2. Materials and Methods

### 2.1. Participants

A total of $N = 30$ children, living and schooled in Italy were tested remotely using the MuLiMi screening web application. Children attended grades two (last months thereof), three, four and five of primary school, covering an age span of about three years. They had been recruited in schools located in different Italian regions that had accepted our invitation to collaborate in the study. $N = 11$ children were monolingual Italian. A total of $n = 19$ children were bilingual.

More precisely, $n = 7$ children spoke Mandarin in addition to Italian, attending an Italian mainstream school, while $n = 12$ children spoke English in addition to Italian and attended bilingual schools where they were schooled in English and Italian. All $n = 11$ monolingual Italian-speaking children also participated in the usability study. Furthermore, $n = 10$ Examiners who had administered the screenings remotely to children (including two of the authors, M.E. and G.M.) completed the online questionnaire on usability.

### 2.2. The MuLiMi Screening Platform

From a technical perspective, MuLiMi is a three-tier, RESTful system, developed following the MVC protocol. As anticipated in Section 1.2, MuLiMi provides different functionalities and interfaces for the three different categories of end-users: Examiner, Examinee and Administrator. Through the Examiner interface, the user can manage the personal data of their Examinees, play tests (either locally or remotely) and analyze the test results. The interface to launch a screening session is shown in Figure 1. Selecting an Examinee will enable the tests suited for them. Then, the Inclusion Criteria must be selected according to the Examinee situation. Optionally, the test can be marked as a remote one, and a link to share with the Examinee will be generated.

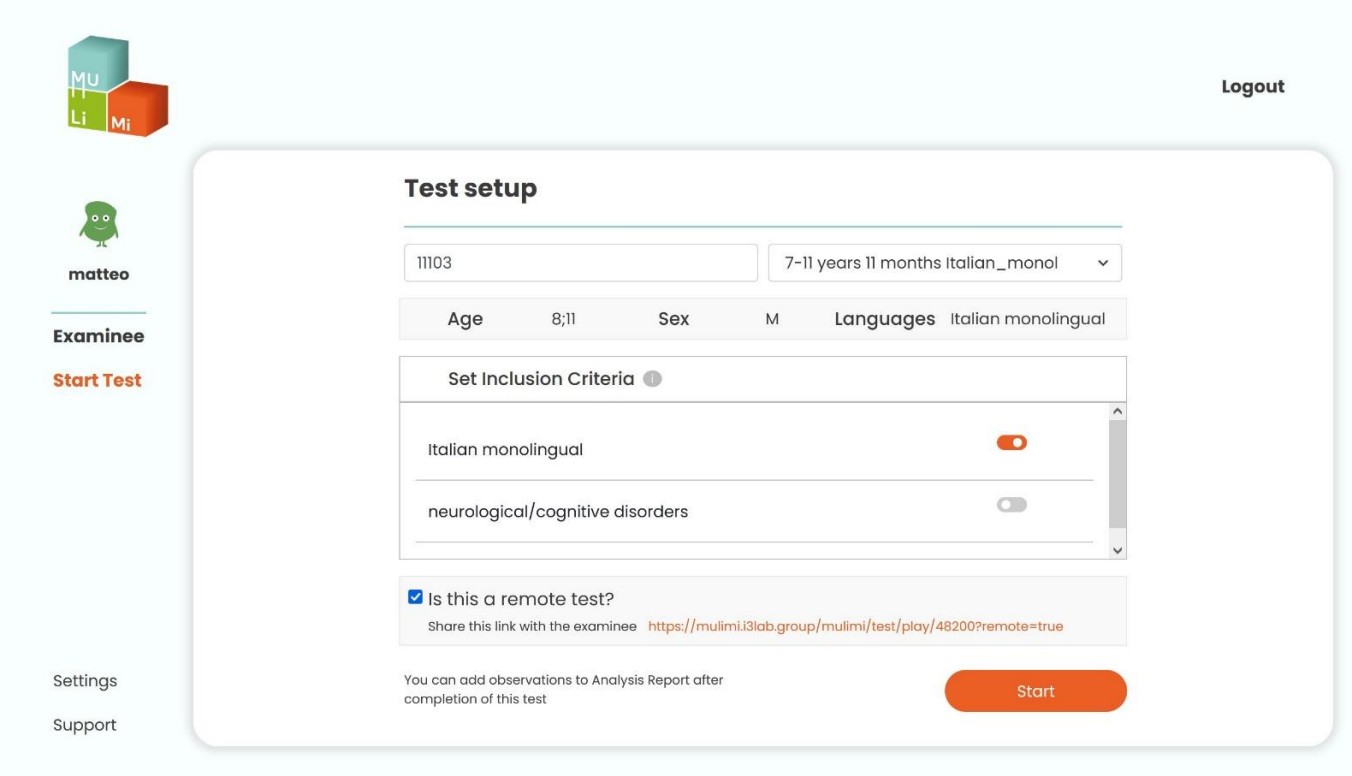

**Figure 1.** The Examiner interface to launch a screening session.

To view the screening results, the Examiner will navigate among all the test sessions undergone by an Examinee and among the Tasks of individual test sessions. The Examiner can also download a spreadsheet containing all the answers recorded during a test session, write remarks for the entire session or specific for single tasks and visualize online aggregated data about the Examinee's performance in each task. The interface for data analysis is shown in Figure 2. The displayed data are:

- The number of correct answers.
- The number of wrong answers.
- The total reaction time (in ms).
- The average reaction time, excluding the fastest and slowest ones to reduce leverage.

The Examinee's interface was designed considering their young age, ranging from 7 to 10 years. Each task type is rendered so that, using adequate multimedia elements (Contents), the experience for the child will be joyful and gamified (see Figure 3 as an example). Another key aspect to improve enjoyability is to avoid boredom as the test goes on. This is achieved by inserting rewarding visual feedback in each of the instructions at the beginning of each task, which provides a reward for the child's progression. A typical testing session starts with a Welcome page. Then, each task in the test is presented, preceded by the related instruction page (which is, by itself, configurable—using text, audio and video Contents—by the Administrators).

The Administrator interface enables the creation of new tests or the configuration of an existing one by supporting a standard workflow and facilitating the reuse of multimedia contents.

The workflow consists of the following steps: an Administrator first creates the Contents, which are primitives representing multimedia files (Figure 4). Supported file types include audio, pictures, videos, plain text and Boolean values.

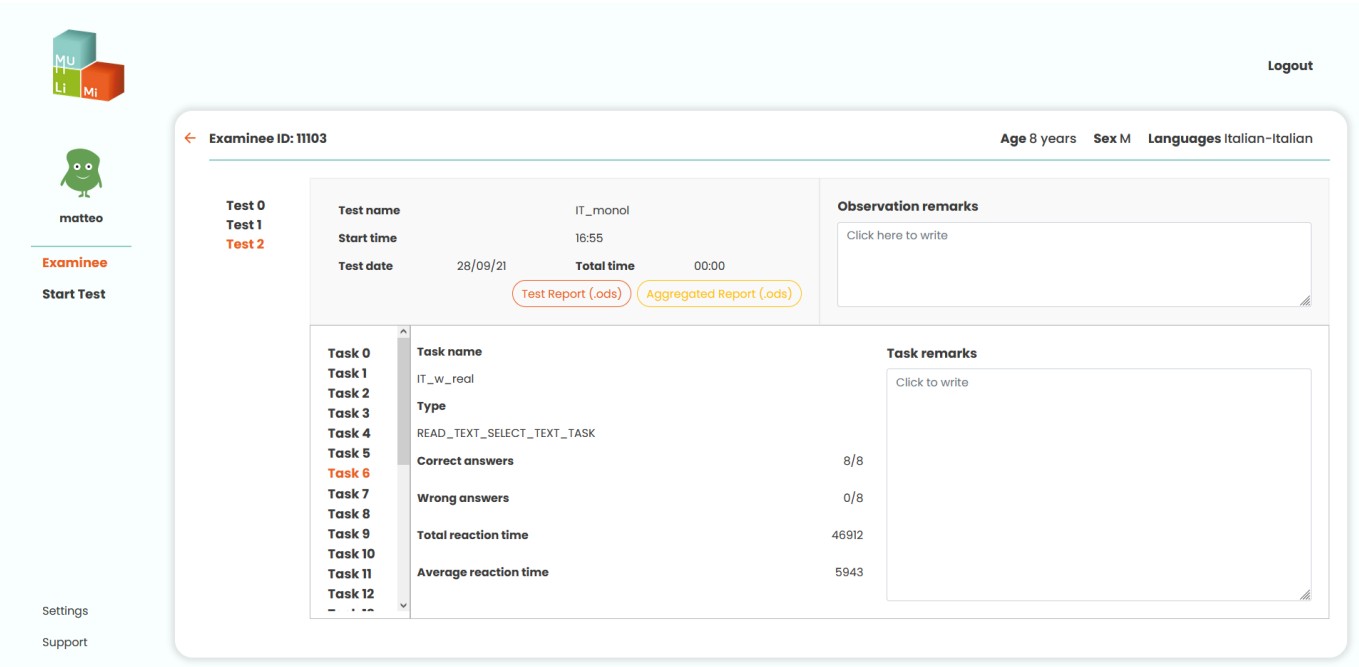

**Figure 2.** The Examiner interface for data review.

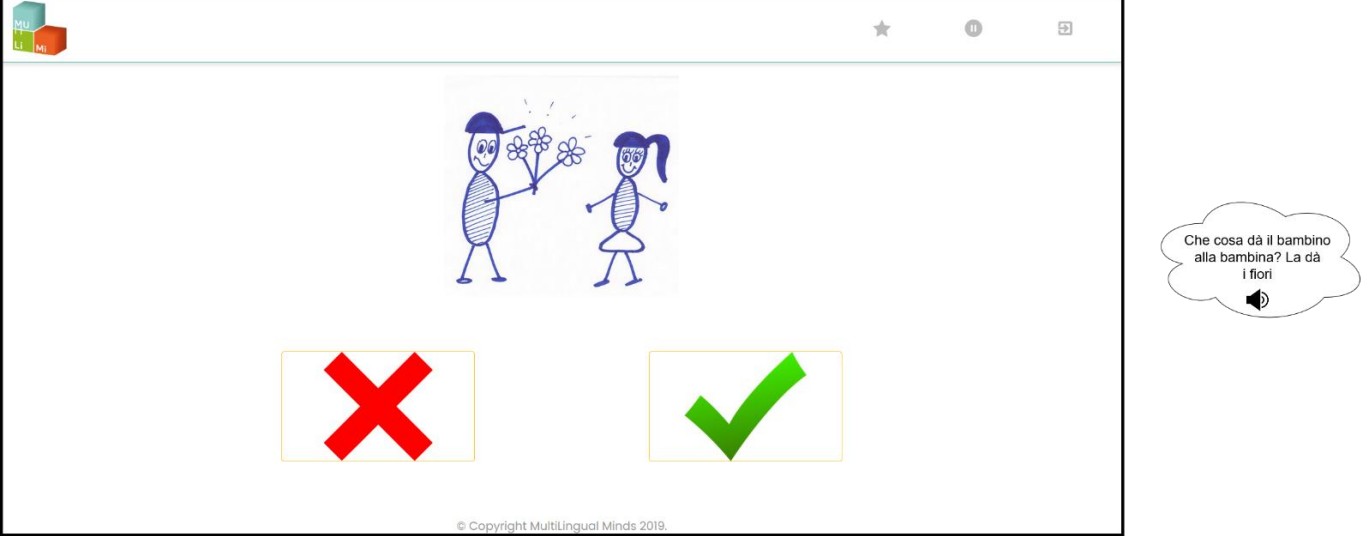

**Figure 3.** The Examiner interface during a Grammaticality Judgment Task showing the simultaneous presentation of different multimedia stimuli relating to the sentence to be judged.

Contents are then used to create Items: they represent individual screens rendered during the testing session. Items are specific to one task type and can be used to compose Tasks of that type only, as the task type changes the semantic meaning conveyed by the contents inside the item (and consequently the logic used to render them).

A Task is a sequence of Items of the same type, preceded by an instruction page (which is provided in written form but additionally read aloud by a recorded voice and whose purpose is to explain to the Examinee what they will have to do during the task).

A sequence of Tasks can finally be combined to form a Test (Figure 5), which represents what will be played during an entire Session. A Test is specific to a Language group (pair of L1 and L2) and an Age group (an interval of ages), and Examinees will perform a screening for risk identification of a language or reading disorder.

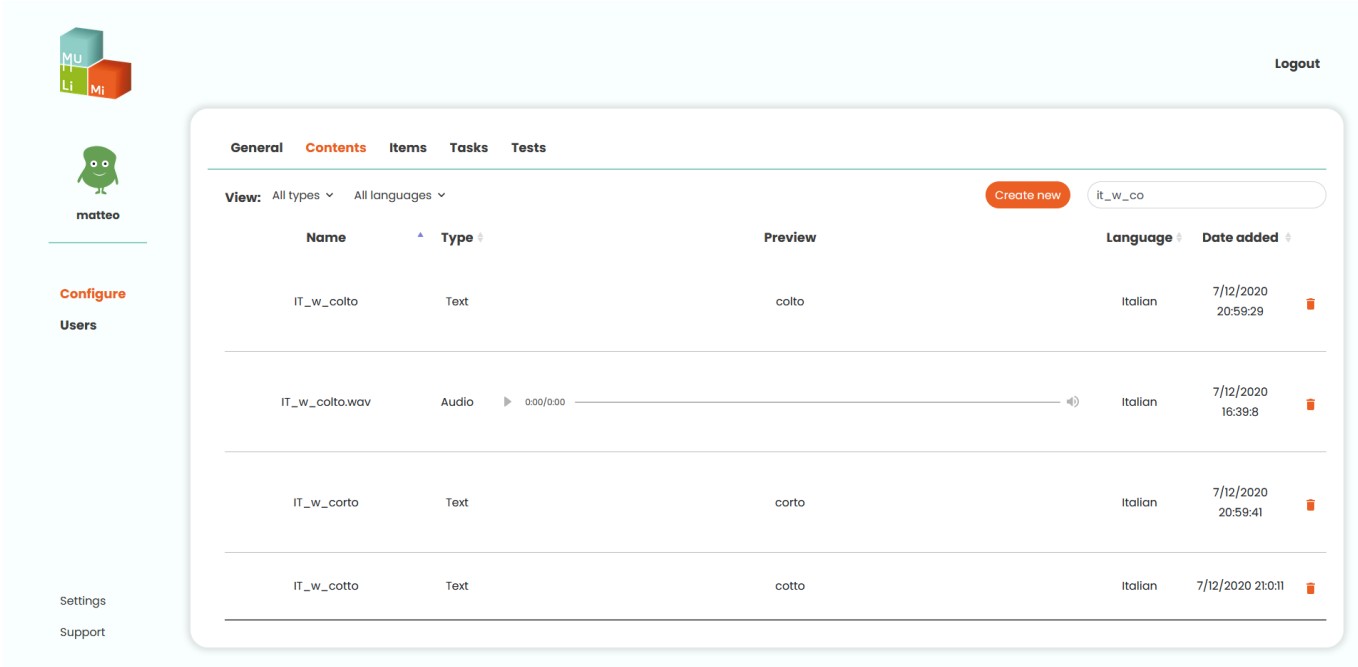

**Figure 4.** The Examiner interface displaying the Content Overview.

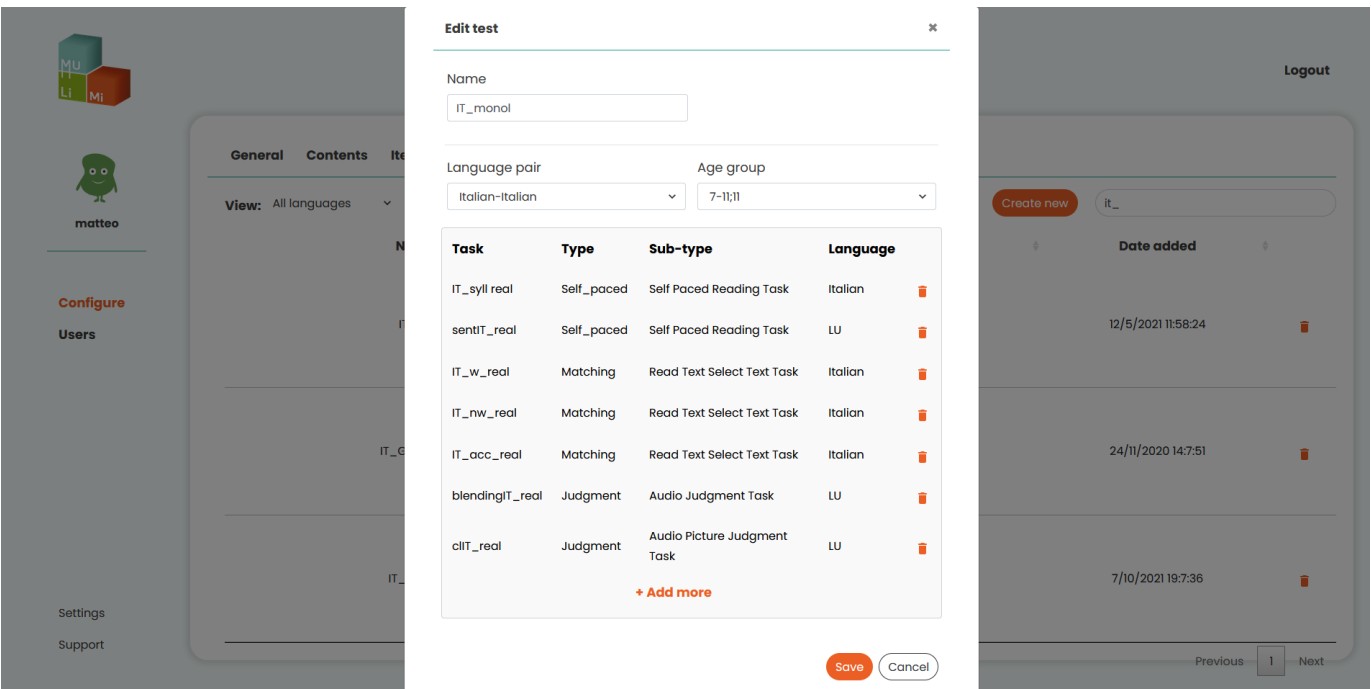

**Figure 5.** The Examiner interface for the creation of screening tests by the insertion of Tasks.

Tasks, Items and Contents can specify a target language. The possible values are dynamically selected among all the individual languages used to compose the Language groups stored into the platform, plus the Language Universal (LU) value. If a specific language is selected, the element can be used only into other elements of the same language, or LU. If LU is selected, the element can be used in any other element. Tasks can be used in Tests having their language present as either mother or societal language or in any Test if their language is LU.

MuLiMi Remote Testing

The platform also allows Examiners to perform tests remotely. These types of tests are identical to normal tests, considering tasks to be executed. What changes is the setting in which the test is performed since the Examiner is not physically with the Examinee. A connection is therefore required between the two devices used by the Examiner and the Examinee so that the former can always know how the user's test is proceeding and can help him. The connection is established exploiting WebRTC to instantiate a peer-to-peer communication between the Examiner's and the Examinee's machines, allowing for real-time screen sharing and voice communication.

When the Examiner prepares the test to be run, they can specify to execute a remote test (Figure 1). The platform will generate a link for the Examinee. Once the Examinee opens the link and allows screen sharing, the platform will establish the connection between the devices. Once the connection is established, the Examiner will be able to see the screen of the Examinee streamed on their own screen (Figure 6).

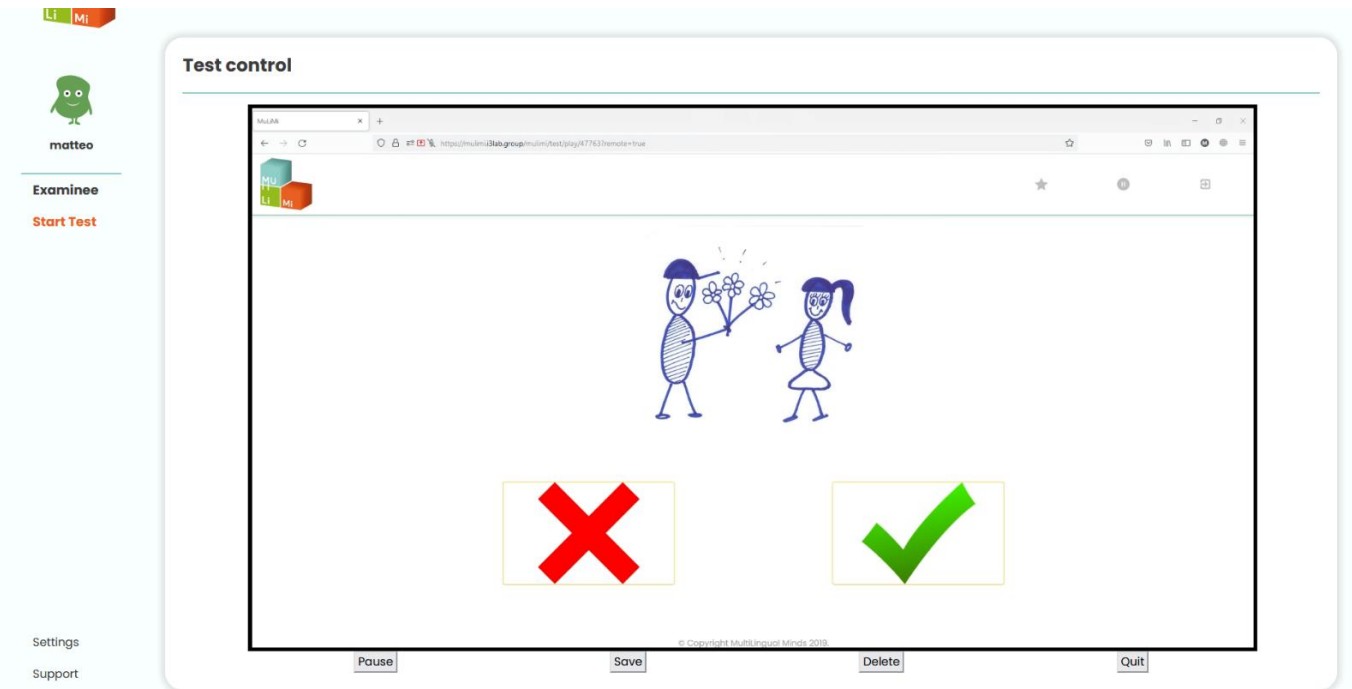

**Figure 6.** The Examiner interface displaying the Examinee's shared screen during remote screening session with an example from a Grammaticality Judgment Task.

### 2.3. Italian Screening Tasks

The Italian Screening Tasks was developed by some of the authors and their collaborators for a preliminary study on Chinese–Italian bilingual children [28]. They have not yet been published nor standardized.

#### 2.3.1. Reading Speed and Accuracy

*Syllable Reading.* The participants were required to read aloud as fast as they could a list of thirty (plus three for training) Consonant-Vowel syllables as in the example: "ba". Syllables were presented one by one on the PC screen. The response/reading time (time elapsed between the presentations of two subsequent syllables) was automatically recorded (self-paced presentation by pressing the spacing bar). The accuracy was not tracked, but it was generally at-ceiling in pilot studies with typically developing children.

*Sentence Reading.* The participants were required to read aloud as fast as they could a list of five sentences (after one training sentence), presented one by one on the PC screen. The sentences increased in length and syntactic complexity, as in the following examples:

"La farfalla vola sui fiori colorati" [the butterfly flies on the colored flowers]; "I gatti camminano lenti sul tetto del palazzo" [the cats walk slowly on the roof of the building]. Sentences were presented one by one on the PC screen. The response/reading time (time elapsed between the presentations of two subsequent sentences) was automatically recorded (self-paced presentation by pressing the spacing bar). The accuracy was not tracked, but it was generally at-ceiling in pilot studies with typically developing children.

*Word Identification* The participants were required to listen to an Italian pre-recorded word (natural voice, native speaker). Then, they had to select the right orthographic form among three visually presented stimuli on the PC screen. Two of those options were distractors (one phonological and one visual distractor). One grapheme was substituted with respect to the target in either case.

For example, they listened to the word "colto" ['colto] [educated] and they had to select the correct orthographic form among the following: "colto" (target), "corto" [short] (phonological distractor) and "cotto" [cooked] (visual distractor). Eight items were presented after a brief training with three items. The response/reading time and accuracy were automatically measured. All targets and distractors were existing words.

*Nonword Identification.* The participants were required to listen to an Italian nonword (pre-recorded, natural voice and native speaker) and, as before, they had to select the correct orthographic form among three visually presented stimuli on the PC screen. Again, two of the options were distractors (one phonological and one orthographic distractor). For example, they listened to the nonword "penko", and they had to select the correct orthographic form among the following: "penco" (target), "benco" (phonological distractor) and "pencio" (pronounced as "pencho", orthographic distractor). Eight items were presented after a brief training with three items. The response/reading time and accuracy were automatically measured.

### 2.3.2. Phonological Awareness

*Stressed Syllable Identification.* The participants listened to an Italian three-syllabic word (pre-recorded, natural voice and native speaker). Then, the word was presented visually and segmented into syllables on the PC screen. They had to identify the stressed syllable among the three presented.

For example, they listened to the word "favola" ['favola; "fairy tale"] and they had to select the stressed syllable among the following visually presented "FA", "VO" and "LA" (the target was "FA"). Eight items were presented after a brief training with two items. Response/reading time and accuracy were automatically measured.

*Phonological Blending.* The participants listened to two audio stimuli (pre-recorded, natural voice and native speaker). The first one consisted of a series of phonemes, presented at a one-second rate. The second audio stimulus consisted of a whole word. Participants were invited to judge if the word (second audio) represented the correct result of the blended phonemes of the first audio or not, by pressing the corresponding buttons displayed on the screen (a green ✓ standing for correct and a red ✖ for incorrect phonological blending).

For example, they listened to the phoneme-by-phoneme segmented word "a-p-r-e"/a//p//r//e/("he/she opens"), and then they listened to the word "arpe" ['arpe] ("harp"); in this case, the blending is incorrect. Ten items were presented after a brief training with two items. A total of 50% of the items were correct, and the other ones contained an inversion of the phonemic sequence. All the targets and distractors were existing words. The response time and accuracy were automatically measured.

*Syllabic Inversion.* The participants were required to listen to two audio stimuli (pre-recorded, natural voice and native speaker). The first stimulus consisted of a bisyllabic word. The second stimulus consisted of the inversion of the two syllables. Participants were invited to judge if the inversion was correct or not by pressing the corresponding correct/incorrect buttons displayed on the screen.

For example, children listened to the word "dado" ['dado], meaning "cube", and then they listened to the correct sequence "doda" ['doda] (resulting from the inversion of the

two syllables) or to the incorrect sequence "donda" ['donda] (both non-existing words). Ten items were presented after a brief training with two items. A total of 50% of the items were correct; the other items contained a violation of the graphemic/phonemic sequence. The response time and accuracy were automatically measured.

### 2.3.3. Grammatical Measures

*Subject–Verb Agreement.* The participants were required to listen to recorded sentences (pre-recorded, natural voice and native speaker) presented one by one and to judge their grammaticality by pressing the corresponding correct/incorrect buttons displayed on the screen. The sentences were taken from a previous experiment on morpho-syntactic processing in DD [41].

Violations consisted of incorrect subject–verb agreement or incorrect auxiliary selection, as in the following examples: "Le galline grasse mangia* sul prato" [the fat hens eat* (singular) on the lawn] is incorrect because the required agreement between the Subject "Le galline grasse" (plural) and the Verb inflection "mangia" (singular) is not realized; the latter ought to be "mangiano" (plural). Ten items were presented (50% correct) after a brief training with two items. The response time and accuracy were automatically measured.

*Clitic Pronouns.* The participants were presented with line drawings shown on the PC screens, and they listened to questions and to the related answers containing a clitic pronoun. All questions and answers were pre-recorded (natural voice, native speaker). The picture remained on the PC screen until a response was provided. They were invited to judge the grammaticality of the answers by pressing the corresponding correct/incorrect buttons displayed on the screen. The violations consisted of the occurrence of an incorrect clitic pronoun.

For example, in the sentence "Che cosa fa il bambino alla bambina? La* dà i fiori" [What is the boy doing to the girl? He is giving her* flowers], the clitic pronoun "la" is incorrect, because the correct occurrence must be the dative-feminine clitic "le", instead of the accusative-feminine "la". Twelve items were presented (50% correct) after a brief training with one item. The response time and accuracy were automatically measured (see Figure 3 in the Examinee and Figure 6 in the Examiner interface).

### 2.4. Mandarin Screening Tasks

The Mandarin screening tasks consist of a Rapid Automatized Naming (RAN) test, two tasks on judgement of correctness of Chinese characters (Hu, unpublished) and three subtests investigating metaphonological skills [48].

*Rapid Automatized Naming (RAN).* A series of Arabic numerals were presented in the center of the screen. The participant is asked to read aloud the number that appears as quickly as possible in Chinese and to press the spacebar on the keyboard to see the next number. In total, the test consists of 25 Arabic numerals (preceded by five training items), representing the random repetition of five numbers used to construct this test (2, 4, 6, 7 and 9). The response/reading time was automatically measured (self-paced presentation by pressing the spacing bar). The accuracy was not tracked, but it was at ceiling in pilot studies.

*Radical Position.* This task examines children's ability to recognize correctness of the position of a radical (visual element) within a Chinese character [49,50]. One character at a time appears on the screen, and the child indicates whether they are presented correctly or not by pressing the corresponding correct/incorrect buttons displayed on the screen. Violations are obtained by changing the position of the radical in the character. The test consists of a total of 18 test items, preceded by two training items. The response/reading time and accuracy were automatically measured.

*Left–Right Inversion.* This task assesses the child's ability to identify the correct orientation of the components of certain characters with a high frequency of use [50]. One character at a time appears on the screen, and the child indicates whether it is spelled correctly or not by pressing the corresponding correct/incorrect buttons displayed on the screen. Violations

are characterized by mirroring a radical contained in the character. The test consists of a total of 18 test items, preceded by two training items. The response/reading time and accuracy were automatically measured.

*Tone Detection*. Two practice trials and eight experimental trials were included. Each item was a Mandarin syllable. Each trial included four items: three with the same tone and one with a different tone. The participants were required to listen to the four audio stimuli and then select the syllable in which the lexical tone differs from the others. For example, they listened to the syllables huā, tán, luó and lán, and they had to select huā. The response time and accuracy were automatically measured.

*Onset Detection*. Two practice items and 16 experimental items were included. The participants listened to four Mandarin syllables and were asked to indicate among the three syllables, which one sounded different. For instance, among tán, tǐng, téng and luó, the syllables tán, tǐng and téng have the same onset "t". Thus, participants were expected to select luó as the different one. The response time and accuracy were automatically measured.

*Rhyme Detection*. Two practice items and 12 experimental items were included. The participants listened to four Mandarin syllables and were asked to indicate which one sounded different. For instance, among tǎng, dáng, láng and qíng, the syllables tǎng, dáng and láng have the same rhyme "ang". Thus, participants were expected to select qíng as the different one. The response time and accuracy were automatically measured.

*2.5. English Screening Tasks*

*Rapid Automatized Naming (RAN)* For the description of the task, see above in the section of Mandarin screening tasks. English-speaking children were asked to name the digits in English.

*Sentence Reading*. For the description of the task, see above in the section of Italian screening tasks. English-speaking children were asked to read out English sentences.

*Orthographic Form Identification*. Participants listened to a sentence (pre-recorded, natural voice and accent, for example "Every girl will dress up as a witch."). The written word form of the last word of each sentence ("witch") along with two distractors was displayed on the screen.

The children had to identify the word spelled correctly in the context of the sentence they had heard before. While one of the distractors was an existing word that was spelled differently but pronounced alike ("which"), the other was a non-existing word that could be pronounced in the same way as the target ("whitch"). Nine items were presented after a brief training with three items. The response/reading time and accuracy were automatically measured.

*Phonological Form Identification*. Participants were visually presented one sentence at a time containing one word that was highlighted ("In class, sometimes teachers project slideshows", highlighted word: [pro'ject]). At the same time, the below three buttons were displayed. While one of the buttons represented the spoken word form of the highlighted word of the written sentence, the other were distractors (distractor 1: ['project]; distractor 2: [pro'tect], pre-recorded words, natural voice and accent). The children indicated the correctly pronounced version of the highlighted written word presented in the context of a sentence. Eight items were presented after a brief training with three items. The response/reading time and accuracy were automatically measured.

*Stressed Syllable Identification*. For the description of the task, see above in the section of Italian screening tasks. English-speaking children instead were asked to indicate the word stress for bisyllabic English words.

*Sound Deletion*. Participants reacted to the questions like "What word would be left if the /b/-sound was taken away from/block/?" and listened to the two options "bock" (incorrect sound deletion) and "lock" (correct sound deletion, pre-recorded, natural voice and accent). Their task was to indicate which of the two options is the result of the correct

sound deletion as requested by the question. Ten items were presented after a brief training with two items. The response/reading time and accuracy were automatically measured.

*Tense Judgment.* Participants reacted to the question "Who says it right?", which was followed by one sentence in the past tense and one sentence in the present tense. Depending on the context of the sentence, either past or present tense use was correct (example: "Last summer it rained a lot." vs. "Last summer it rains* a lot."). Their task was to indicate which of the two options was correct. Twelve items were presented after a brief training with two items. The response/reading time and accuracy were automatically measured.

### 2.6. Standardized Reading Tests

*Batteria per la Valutazione della Dislessia e della Disortografia Evolutiva-2* (DDE-2 [29]). The DDE contains a word reading subtest including four vertically displayed lists of 28 words and a nonword reading subtest including three lists of 16 nonwords (non-existing words).

*Chinese reading test* (Hu, in preparation). A test including 150 two-character words, which were chosen from the sets of most popular Chinese language textbooks used in Chinese schools in Italy was presented on the screen. One point was assigned when both characters of each word were read correctly, and 0.5 point was assigned when one character of each word was read correctly. In the Chinese reading test, accuracy, but not reading speed was assessed for each single character presented. Characters included in the test increased in difficulty. Norms have not been published yet; therefore, raw scores (percent accuracy) were used.

*Test of Word Reading Efficiency—Second Edition* (TOWRE-2 [30]). Children were presented with vertically displayed word lists and asked them to read out loud as accurately and as fast as possible. For both the word and nonword list, the child had 45 s to read as many items as possible. The Examiner interrupted the child when the 45 s were over.

### 2.7. Usability Questionnaires

In order to collect data on the usability of the screening platform as perceived by the Examiners and Examinees/child participants, questionnaire items that apply to the MuLiMi screening platform were selected from the commonly used usability measurement tools "System Usability Scale" [51], Questionnaire for User Interface Satisfaction (QUIS [52]) and Questionnaire of User Interface Satisfaction (QUIS [53]) and translated into Italian and adapted when necessary.

A short online questionnaire containing twelve questions was designed for the children, investigating their opinion of the screening platform (see full questionnaire Appendix A). Each questionnaire item (to be identified by an initial "P" for Participant) consisted of a statement, which was followed by a five-point scale with two extremes that the child could choose from.

A different version of the online questionnaire was designed to be filled by Examiners, investigating their opinion of the screening platform (see full questionnaire Appendix B). Each questionnaire item (to be identified by an initial "E" for Examiner) consisted of a statement that was followed by a five- to nine-point scale with two extremes that the Examiner could choose from. The questionnaire consisted of 46 questions.

### 2.8. DSA Parent Questionnaire

The "DSA Questionnaire" (Lorusso and Milani, unpublished, where DSA stands for the Italian term "Disturbi Specifici di Apprendimento"—Specific Learning Disorders), used on the Institute's online platform to collect information about the children's problems and orient their clinical diagnostic pathway before admission to the clinic, was used here. In addition to questions on anagraphic data and information on the child's language background, from which the children's language dominance was derived, the questionnaire contained questions in the four different categories covering (1) school discomfort, (2) general learning difficulties and (3) in-depth analyses on learning difficulties for reading and writing acquisition and (4) math skills separately.

One point was assigned whenever parents responded with "yes" to a question related to a negatively connoted question. In a second step, compound scores on the four categories were created, and responses from online and pen-and-paper questionnaires were merged and translated into English for data processing.

*2.9. Procedure*

For bilingual children, standardized and screening tests were administered in two separate sessions lasting 45 to 60 min each, while the monolingual children who only received standardized and screening tasks in Italian (their native language) completed all tasks in a single session of approximately 90 min, with a 10-min break. All the children were tested remotely while the Examiner was connected via video conferencing and administered the standardized tasks through screen share. The screening was started using the remote feature of the MuLiMi screening platform requiring the sharing of the link with the child.

In order to simulate a realistic testing scenario, Examiners used their personal or work PC while children used the PC of the school when being tested at school and their families' private computer when being tested at home. Since children were tested exclusively remotely, the standardized test administration procedure had to be adapted accordingly.

After the Examiner had verified that the size of the scanned version of the word lists was comparable to the original size of the characters in the standardized test, children were asked to read the words displayed on their screen through screen share as accurately and fast as possible row by row. The Examiner measured the reading time and scored the child's reading performance after the test administration based on audio recordings as indicated in the test manual.

Since some collaborating schools had expressed preferences for either on- vs. offline formats of the parental questionnaires and to ensure that all the parents could fill in the questionnaires, several versions of the parental questionnaire were created and distributed. While, for the monolingual Italian and English–Italian children, parental questionnaires were implemented as online surveys using "Google Forms", parents of Chinese–Italian children were administered a pen-and-paper version of the questionnaire. Parents of bilingual children had the option to choose whether they wanted to fill in the questionnaire in their L1 or in Italian. The usability questionnaires were also filled by Examinees/child participants and Examiners using "Google Forms". The link to the online survey was shared with the user. Additionally, a pen-and-paper questionnaire was filled in by the teachers of bilingual children, judging the child's Italian productive and receptive phonology, morphosyntax and vocabulary skills, and his/her reading performance on a 5-point-scale.

*2.10. Data Analysis*

In order to display varying degrees of reading skills irrespectively of the presence of an official diagnosis of DD, based on the results obtained in the DDE-2 and the reading tests in the children's L1, a reading difficulty risk score was created for the bilingual children (for the monolingual children the information on DDE-2 performance is sufficient to identify the risk of DD). For the creation of this score, children were first assigned a point whenever they had z-scores at or below minus two standard deviations in speed and/or accuracy in the word and/or nonword reading subtests of the DDE-2.

The DDE-2 risk scores ranged between 0 (no risk in either of the two subtests), 1 (at or below minus two standard deviations in one of the two subtests) and 2 (at or below minus two standard deviations in the two subtests). In a second step, the risk for reading difficulties in the L1 was assessed. For the English-speaking children, the z-scores from the TOWRE-2 were processed equivalently to the DDE-2. For the Mandarin-speaking children instead, when the children were not able to read more than one-third (33.33%) of the Chinese words, a risk of 1 was assigned. In a third step, the sum of those two scores was created and led into the total compound risk score, ranging from 0 to 4.

In order to explore the associations between scores obtained in the different tests (standardized tests and screening tasks) it was necessary to analyze the results separately

in the different language groups (due to different L1 characteristics), thus, reducing the sample size. In particular for the Mandarin-speaking subgroup, which included only seven children, extreme power reduction led us to the decision to consider correlations (Spearman's rho) larger than 0.5, regardless of their significance level, merely regarding them as an indication of a possible tendency in the results, to be confirmed by future studies on larger samples.

For all analyses conducted to answer the research questions described in the introduction, no Bonferroni correction was applied (a-priori hypotheses). Similarly, no correction was applied when performing correlation analyses with highly inter-correlated variables. Whenever sample size, non-continuous variables or clear deviations from normality prevented application of parametric correlations, Spearman's correlations were computed.

## 3. Results

### 3.1. Standardized Reading Test Results

In a first step, the bilingual children's reading skills in Italian as measured by the DDE-2 were compared to their L1 reading performance measured by the TOWRE-2 and the Mandarin reading test, respectively. The amount of correctly read characters in the Mandarin reading test was associated (but not significantly) with the performance in the word reading subtest of the DDE-2 (accuracy: $n = 7$, $rho = 0.667$, $p = 0.102$, reading time: $n = 7$, $rho = 0.691$, $p = 0.086$). Significant associations were found comparing the English-speaking children's reading performance in the DDE-2 with the reading performance in the TOWRE-2 for reading time and accuracy across word and nonword reading tasks in the two tests ($N = 12$, $rho_s$ between 0.586 to 0.772, $p_s$ ranging from 0.003 to 0.045).

### 3.2. Screening Results

The raw scores obtained in the screening tasks were analyzed for the total group of participants ($N = 30$), first considering the whole sample and then the three different groups (monolingual, Chinese–Italian and English–Italian children) separately. Various parameters were compared to the results in the screening tasks: performance in standardized tests (3.2.1 and 3.2.2), the risk level deriving from the standardized reading test(s) (3.2.3) and the risk factors for impaired reading acquisition assessed through parental questionnaires (3.2.4). Furthermore, results on the usability studies will be described (3.3).

#### 3.2.1. Comparison of Screening and Standardized Test Results in Italian

*Reading Time-related tasks.* The total reading time for words (DDE-2) was significantly associated with both the mean reading time for a syllable ($n = 27$, $r = 0.461$, $p = 0.015$) and the mean reading time for a sentence ($N = 30$, $r = 0.942$, $p < 0.001$) in the self-paced reading paradigms in the screening tasks. Similar results were obtained comparing the total reading time for nonwords (DDE-2) to the mean reading time per item in the self-paced syllable ($n = 27$, $r = 0.406$, $p = 0.036$) and sentence ($N = 30$, $r = 0.942$, $p < 0.001$) reading tasks.

While the time in matching an auditorily presented word to one out of three written words in the screening was significantly associated with both total reading time of words ($N = 30$, $r = 0.531$, $p = 0.003$) and nonwords ($N = 30$, $r = 0.582$, $p = 0.001$) in the DDE-2, no significant associations were found for response time in the audio-nonword-matching task with any of the standardized subtests ($p > 0.05$).

The aforementioned associations were still present when looking at the three subgroups individually, even though due to smaller sample sizes, the significance levels were lower. While Figure 7 shows that on average, slow readers in the DDE word reading subtest (*y*-axis) were also slower in matching an auditorily presented word to one out of three written words, it also becomes evident that all the children from the English-speaking subgroup (red dots) read slower compared to the other groups.

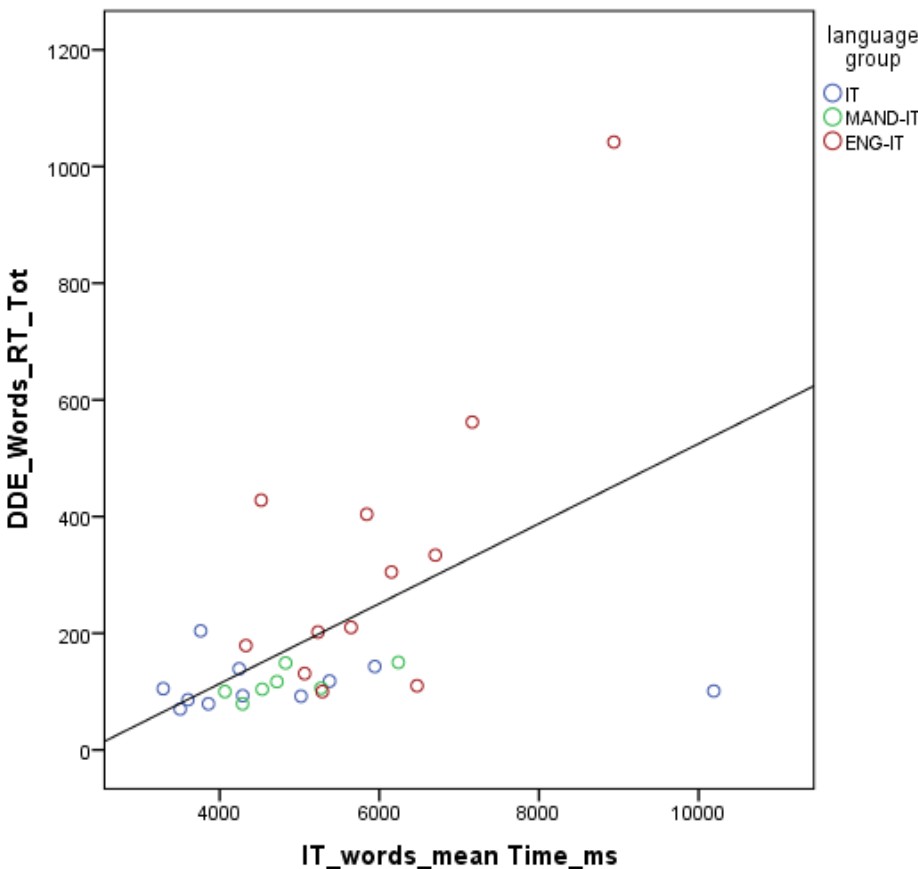

**Figure 7.** Scatterplot relating the Mean Word Identification time (in ms) in the Italian screening task (*x*-axis) and total reading time (in sec) on a standardized word reading tasks (*y*-axis) in the three subgroups (represented by the three different colors). Linear $R^2 = 0.282$.

*Reading Accuracy-related tasks.* Not only the reading time but also the accuracy in the reading screening was significantly associated with accuracy in the standardized reading tests. In particular, there was a significant association between the percentages of correctly read words in the DDE-2 and the percentages of correctly matched audios to one out of three written word forms presented on the screen ($N = 30$, *rho* $= 0.584$, $p = 0.001$). A similar result emerged when comparing nonword reading accuracy in percentages in the DDE-2 to percentages of correctly matched audios to one out of three written nonwords presented on the screen ($N = 30$, *rho* $= 0.410$, $p = 0.025$).

These effects were still present when comparing the reading (DDE-2) and matching performance (MuLiMi screening) of words and nonwords in the three subgroups individually. While the correlations were very strong in the English–Italian speaking subgroup, they were less strong in the other two groups. Figure 8 also highlights again that the weakest readers were part of the English–Italian group.

*Language-related tasks.* The percentages of correctly judged items in the Phonological Blending task in the screening were associated with the accuracy in the word ($N = 30$, *rho* $= 0.492$, $p = 0.006$) and nonword subtest ($N = 30$, *rho* $= 0.356$, $p = 0.053$) of the DDE-2. Similar effects instead were neither found when comparing results of the accent identification task ($n = 27$, $rho_s < 0.333$, $p_s > 0.05$) nor when comparing the results of the standardized reading subtests to the children's performance in the Syllabic Inversion task, which was carried out with the monolingual Italian and Mandarin–Italian-speaking children only ($n = 18$, $rho_s < 0.052$, $p_s > 0.05$).

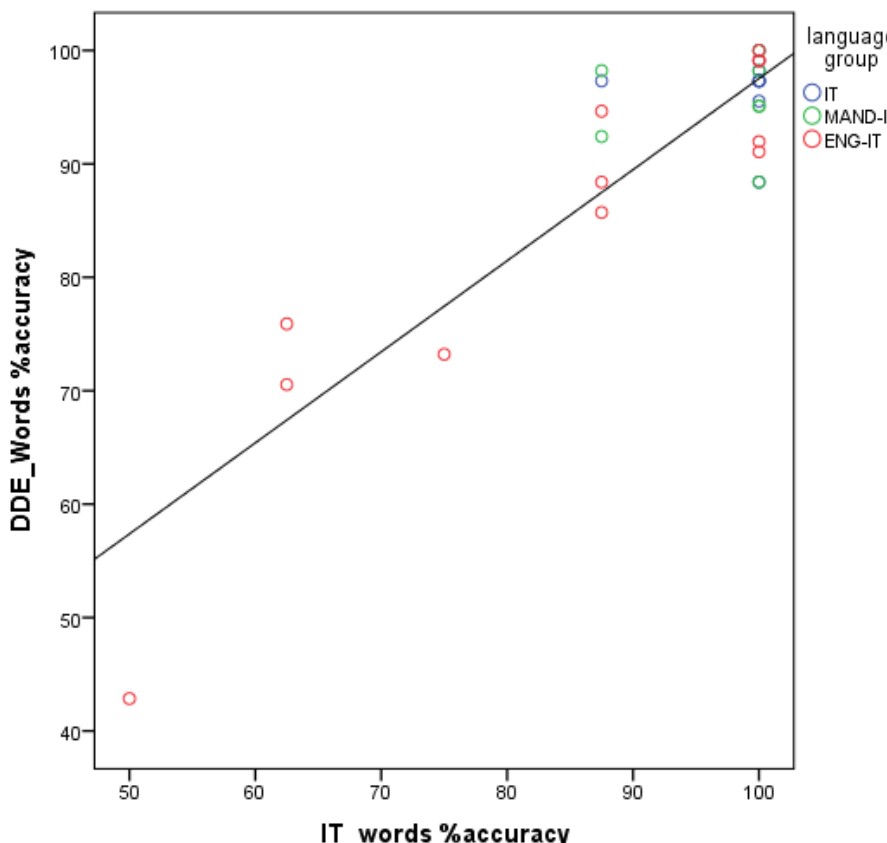

**Figure 8.** Scatterplot relating percent of Word Identification accuracy in the Italian screening task (*x*-axis) and in the standardized reading tasks (*y*-axis) in the three subgroups (represented by the three different colors). Linear $R^2 = 0.791$.

Investigating the three different subgroups individually instead, except for the accuracy in grammaticality judgements of Clitic Pronouns in the group of English-speaking children (*n* = 12, correlations with DDE word reading accuracy: *rho* = 0.713, *p* = 0.009, nonword reading accuracy: *rho* = 0.676, *p* = 0.016), for none of the aforementioned comparisons did we find significant associations ($rho_s < 0.658$, $p_s > 0.05$). Furthermore, the performance of judgement of correctness of subject–verb agreement, which was carried out with the monolingual Italian and Mandarin–Italian-speaking children only, was not associated with the reading accuracy in the standardized reading tests (*N* = 18, *p* > 0.05).

While the performance in judgement of clitic object use in sentences was significantly associated with reading accuracy in the DDE-2 word reading task (*N* = 30, *rho* = 0.513, *p* = 0.004), it was not associated with the performance in the nonword reading subtest (*N* = 30, *p* > 0.05). When investigating the three different subgroups individually, significant associations emerged in the English–Italian subgroup only for DDE word (*n* = 12, *rho* = 0.713, *p* = 0.009) and nonword (*n* = 12, *rho* = 0.676, *p* = 0.016) reading accuracy.

3.2.2. Comparison of Screening Tasks and Standardized Test Results in the Child's L1

*English–Italian screening.* Reading time, measured automatically through the self-paced sentence reading screening task, was significantly associated with the reading time for words (*n* = 12, *rho* = 0.888, *p* < 0.001) and nonwords (*rho* = 0.690, *p* = 0.013) in the TOWRE-2. The reaction time in the Orthographic Form Identification task was significantly associated with the reading time for words (*rho* = 0.713, *p* = 0.009) and nonwords (*rho* = 0.701, *p* = 0.011) in the TOWRE-2.

Investigating the accuracy of the children's performance in the standardized and screening tasks instead, the percentage of correctly read words in the TOWRE-2 was signif-

icantly associated with the percentage of correctly identified words in the Orthographic Form Identification in the screening ($rho = 0.748$, $p = 0.005$).

While no significant association was found for accuracy between the TOWRE-2 and phonological awareness screening tasks ($rho_s > 0.481$, $p < 0.05$), significant associations were found for the accuracy in the English grammaticality judgement task on tense marking and word reading accuracy ($rho = 740$, $p = 0.006$). Furthermore, several associations were found between the reading time and accuracy in English screening tasks and the Italian standardized reading tasks ($rho_s$ between 0.583 and 0.895, $p_s < 0.037$).

*Chinese–Italian-screening.* The Mandarin screening task on Onset Detection was significantly associated with the amount of characters correctly read in the Chinese word reading test ($N = 7$, $rho = 0.873$, $p = 0.010$). Further associations (non-significant) emerged comparing the percentage of correctly read characters in the Chinese word reading test with the accuracy in the Left–Right Inversion judgement ($rho = 0.574$, $p = 0.178$), Radical Position judgement ($rho = 0.574$, $p = 0.178$) and Rhyme Detection task ($rho = -0.625$, $p = 0.133$).

### 3.2.3. Associations between the Screening Task Results and Risk Level

For the bilingual groups only ($N = 19$), risk scores were compared to the performance in Italian and L1 screening tasks. The L1 risk score was significantly associated with the mean reading time in the self-paced sentence reading task ($rho = 0.473$, $p = 0.041$) and accuracy in the Word Identification task ($rho = -0.500$, $p = 0.029$), the compound risk score was associated (but not significantly) with the self-paced sentence reading time ($rho = 0.428$, $p = 0.068$) and accuracy in the Nonword Identification screening task ($rho = -0.406$, $p = 0.085$).

A significant association instead emerged between the compound risk score and the mean response time in the Word Identification screening task ($rho = 0.457$, $p = 0.049$) and (a negative one) the time in the judgement of correctly or incorrectly used Clitic Pronouns ($rho = -0.531$, $p = 0.019$).

Also considering the two bilingual groups separately, significant associations between the risk scores and performance in the L1 screening tasks emerged. The mean reading time in the English self-paced sentence reading screening task was associated with the risk scores (L1 risk derived from the z-scores in the TOWRE-2: $N = 12$, $rho = 0.717$, $p = 0.009$, compound risk score: $N = 12$, $p = 0.566$, $rho = 0.055$). The L1 risk score was also significantly associated with accuracy in the English Orthographic Form Identification screening task ($rho = -0.570$, $p = 0.053$) and accuracy in the grammaticality judgement on tense marking ($rho = -0.781$, $p = 0.003$). Further associations with the L1 risk score (though not significant) emerged with response time in the English pronunciation identification task ($rho = -0.512$, $p = 0.089$) as well as accuracy ($rho = -0.493$, $p = 0.104$) and response time ($rho = 0.512$, $p = 0.089$) in the English deletion judgement task.

As for the Mandarin-speaking children, performance in the L1 screening tasks was significantly associated with the compound risk score. In particular, accuracy in the Left–right Inversion judgement ($N = 7$, $rho = -0.882$, $p = 0.009$) and in the Radical Position ($rho = -0.882$, $p = 0.009$) as well as response time in the Onset Detection task ($rho = -0.866$, $p = 0.012$) were significantly associated with the compound risk score. Further associations (non-significant) emerged analyzing the response time in the judgement of Mandarin Radical Position, Rhyme Detection, Tone Detection and RAN (for each one, $N = 7$, $rho = -0.577$, $p = 0.175$). The pattern was confirmed when comparing the performance in the aforementioned screening tasks to the L1 risk score.

### 3.2.4. Associations between Screening Results and Parental and Teachers' Questionnaires

The compound scores based on the answers to the questions in the section on reading and writing acquisition in the parental questionnaire were significantly associated with the reading time in the self-paced sentence reading screening task ($N = 30$, $rho = 0.568$, $p = 0.001$), response time and accuracy in Italian word (accuracy: $rho = -0.625$, $p < 0.001$;

time: rho = 0.399, $p$ = 0.029) and Nonword Identification (accuracy: $rho$ = −0.477, $p$ = 0.008, time: $rho$ = 0.426, $p$ = 0.019).

These scores were also significantly associated with both the response time ($rho$ = 0.580, $p$ = 0.001) and accuracy ($rho$ = −0.427, $p$ = 0.018) in the Italian blending judgement and accuracy in the Italian agreement ($rho$ = −0.728, $p$ = 0.001) and clitic judgement ($N$ = 30, $rho$ = −0.482, $p$ = 0.007) tasks. For the compound score of the section "school discomfort", significant association emerged with accuracy on Italian Word Identification ($rho$ = −0.372, $p$ = 0.043). The compound score on general learning difficulties instead was significantly associated with accuracy in the Italian Nonword Identification screening task ($rho$ = −0.371, $p$ = 0.044) and with response time in the grammaticality judgement of clitic pronoun use ($rho$ = 0.384, $p$ = 0.036).

Interestingly, the compound score of the section of mathematical problems of the parental questionnaire was also significantly associated with accuracy in the Italian Word Identification judgement screening task ($rho$ = −0.535, $p$ = 0.002). This general pattern of results was still present when looking at the three subgroups individually.

The children's accuracy in the Italian Word Identification task was significantly associated with the teachers' judgements on children's phonological receptive ($N$ = 17, $rho$ = −0.482, $p$ = 0.050) and productive phonological skills ($rho$ = −0.482, $p$ = 0.050) as well as with receptive morphological ($rho$ = −0.494, $p$ = 0.044) and receptive vocabulary skills ($rho$ = −0.494, $p$ = 0.044).

### 3.3. Usability

Descriptive statistics are used in order to describe the general trend of opinion on user experience and response patterns.

### 3.3.1. Usability Examinees/Participants

In order to facilitate the interpretation of the children's responses, the scores for question P1 were inverted so that a higher score was always representative of a positive user experience. The results of the usability study showed that overall, the participants' experience with the screening platform was positive (see Figure 9) On average, children declared that they were able to use the software autonomously (P1), that it was not difficult to learn to use the system (P4) and that it was simple to use the system (P6).

Furthermore, they described their user experience as generally pleasant when being asked about how they liked the software (P2) and whether it was boring, which the majority did not find, but individual respondents did (see Figure 9, P3). In P7, children expressed that they were generally comfortable in using the system. While the participants expressed satisfaction regarding the readability of characters on the screen (P9), the ability to track the cursor position (P11) and the screen graphics (P12), children did not always find it easy to select screen elements (P9) and also had varying opinions on the feedback provided on whether an element was selected (P11). The children were also generally unsatisfied with the system speed (P5).

The children's opinion on ease of learning to use the system (P4) was significantly associated with their age ($N$ = 11, $rho$ = 0.807, $p$ = 0.003). Age was not significantly associated with any of the other responses the children gave (all $rho_s$ < 0.445, $p_s$ > 0.05). In post-hoc analyses aimed to understand the reasons behind the children's responses, it was found that the children's opinion on the ease of using the system (P6) was significantly associated with their overall reaction to the software (P2 "terrible—wonderful": rho = 0.634, $p$ = 0.036); P3 ("dull—stimulating": $rho$ = 0.926, $p$ < 0.001).

Children's impressions on how comfortable they felt using the software (P7) was significantly associated with how boring (P3, $rho$ = 0.898, $p$ < 0.001) and how simple (P6, $rho$ = 0.946, $p$ < 0.001) they perceived the system. The ability to see the cursor position (P10) was almost significantly associated with ease in learning how to use the system (P4, $rho$ = 0.588, $p$ = 0.057).

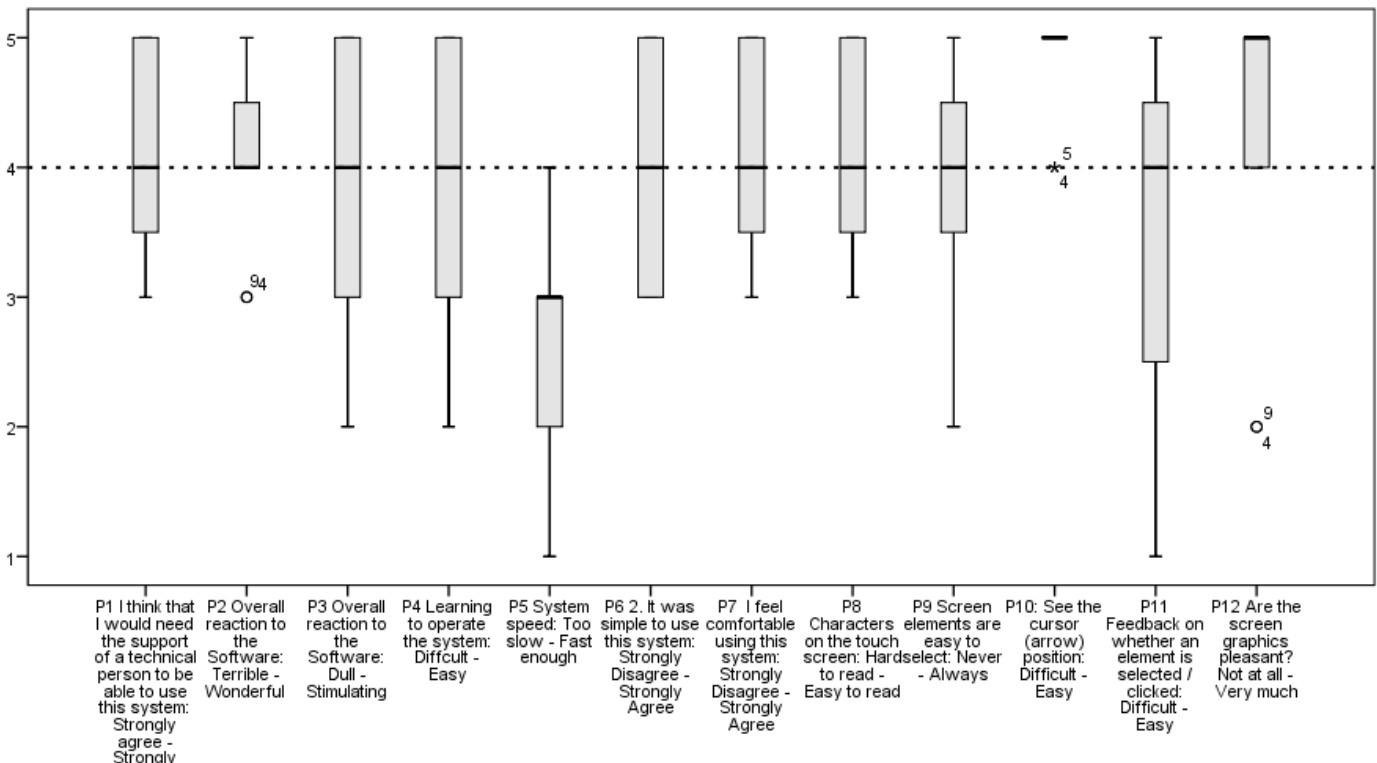

**Figure 9.** Boxplots representing the distribution of Examinee/child participant opinions on the screening platform expressed on a five-point scale. For each question, the black horizontal line inside the box represents the median score expressed, and the grey box represents the extension from the first to the third quartile. The dotted line represents the median score for the whole scale.

### 3.3.2. Usability Examiners

Again, in order to facilitate the interpretation of the responses, the response options of questions E2, E4, E7, E9 and E38 were inverted so that a higher score was always representative of a positive user experience. Figure 10 represents the Examiners' responses on a five point scale to questions taken from the System Usability Scale [51] in boxplots. While the respondents agreed that the software as not too complex to use (E2) and that they did not need to learn a large amount before using the system (E9), responses on the need of support by a technician (E4), confidence in using the system (E8) and the integration of functions in the system (E5) were more wide-spread across the whole scale ranging from one (negative impression) to five (positive impressions).

Even though the mean value for the responses regarding the ease of use of the system (E3) was still in the upper half of the five-point scale), the reactions to this question were more negative compared to the other questions. While the Examiners' impression of cumbersomeness varied (E8), there was a good degree of agreement on the impression that most people would be able to learn to use the system quickly (E6).

Turning to the overall reaction to the software (QUIS [52], see Figure 11 for an overview), while judgement on the system's flexibility was rather negative (E14), the raters expressed positive overall experience (E10) and did not find its application boring (E13). Average impressions on the ease of the software use (E11) and the general satisfaction with the software (E12) were similar.

Further questionnaire items adapted from the QUIS [52] more specifically targeted the screen layout. Both, the deciphering of characters on the screen (E15) as well as the organization of the screens (E16) were considered positive apart from the two outliers (see Figure 12). While the users considered that it was easy to learn how to operate the system

(E17) and rather straightforward to perform the tasks (E18), users were less confident about the system speed (E19), reliability (E20) and suitability for all levels of users (E21).

**Figure 10.** Boxplots representing the distribution of Examiner opinions on the screening platform (E2–E9). For each question, the black horizontal line inside the box represents the median score expressed, and the grey box represents the extension from the first to the third quartile. The dotted line represents the median score for the whole scale.

Further questionnaire items assessed the perceived capability of the system to meet the screening requirements as perceived by the Examiners (see Figure 13). Overall, they expressed personal satisfaction (E26) and satisfaction regarding the functionalities and the design of the platform (E23, E24). While they do generally think that the application of the platform is useful in their work context (E22), some of the respondents were missing functions that they expected (E25).

For further online questionnaire items on graphics, the respondents overall reported positive user experiences (see Figure 14). They were satisfied with the resolution (E27) and the font of characters (E28). While again satisfaction was expressed regarding the contrast with the background (E29), the respondent opinions were less univocally positive for the facilitation of task through the organization of screen layouts (E32) and highlighting (E30). Some of the respondents doubted the helpfulness of the colors used for the highlighting (E31).

While the amount of information on the screen appeared rather adequate to most of the respondents (E33), some of the respondents considered the arrangement of the information illogical (E34). Questionnaire items E35 to E40 regard the selection of items on the screen and the feedback processes involved. While, for the identification (E36) and selection process (E35) as well as the selection area size (E37), the respondents were rather satisfied, the respondents highlighted the need for improvement regarding the system's feedback

provision on cursor location (E38), successful selection of an item (E39) and reliability in responding to the selection (E40).

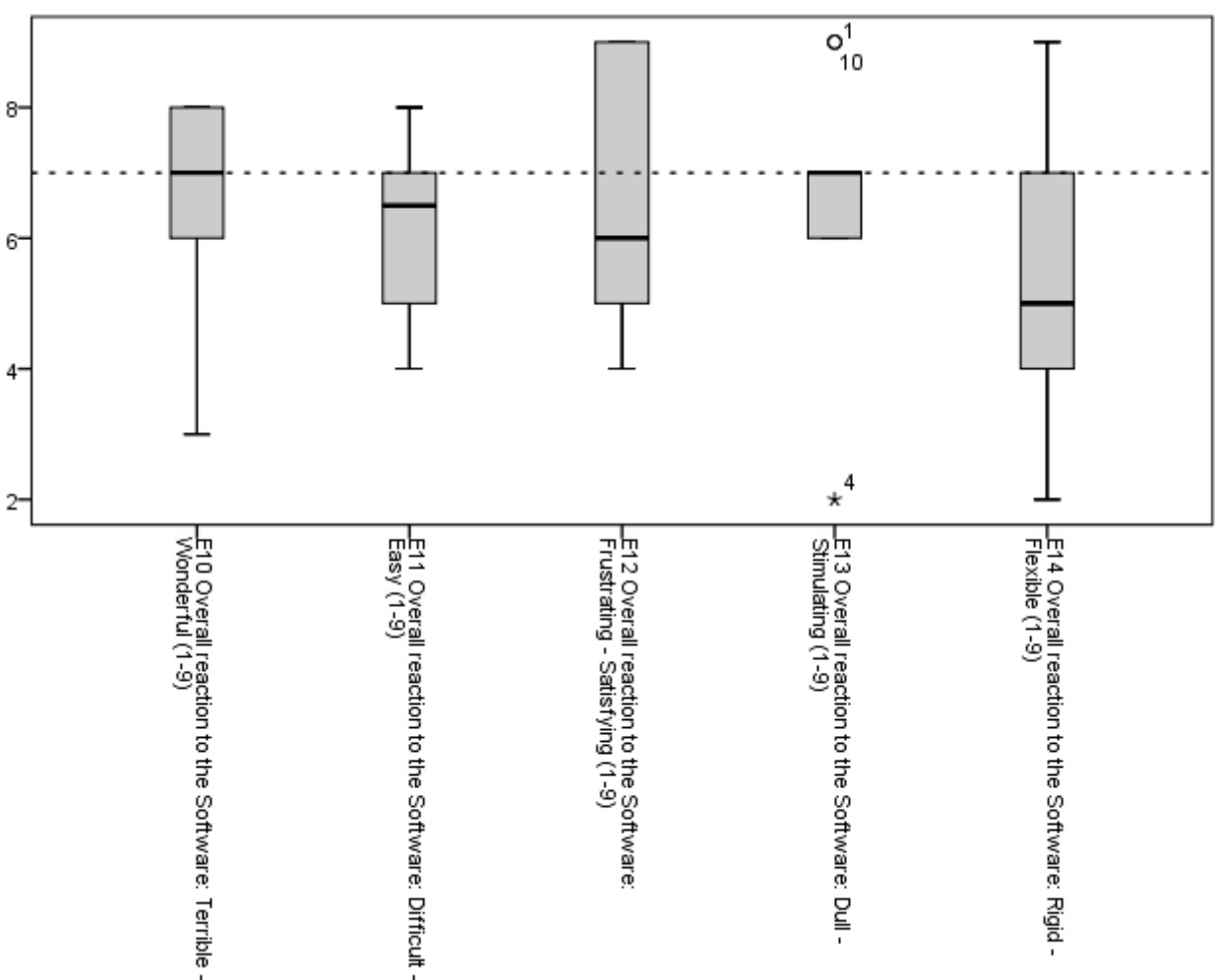

**Figure 11.** Boxplots representing the distribution of Examiner opinions on the screening platform (E10–E14). For each question, the black horizontal line inside the box represents the median score expressed, and the grey box represents the extension from the first to the third quartile. The dotted line represents the median score for the whole scale.

A similar response pattern is revealed on the question on overall system response time of the system (E43). While moderate responses occurred relating to the ease (E41) and time (E42) required to learn to operate the system, respondents were pleased with the screen designs (E44) and the use of colors (E45). Most of the participants had a positive overall user experience (E 46).

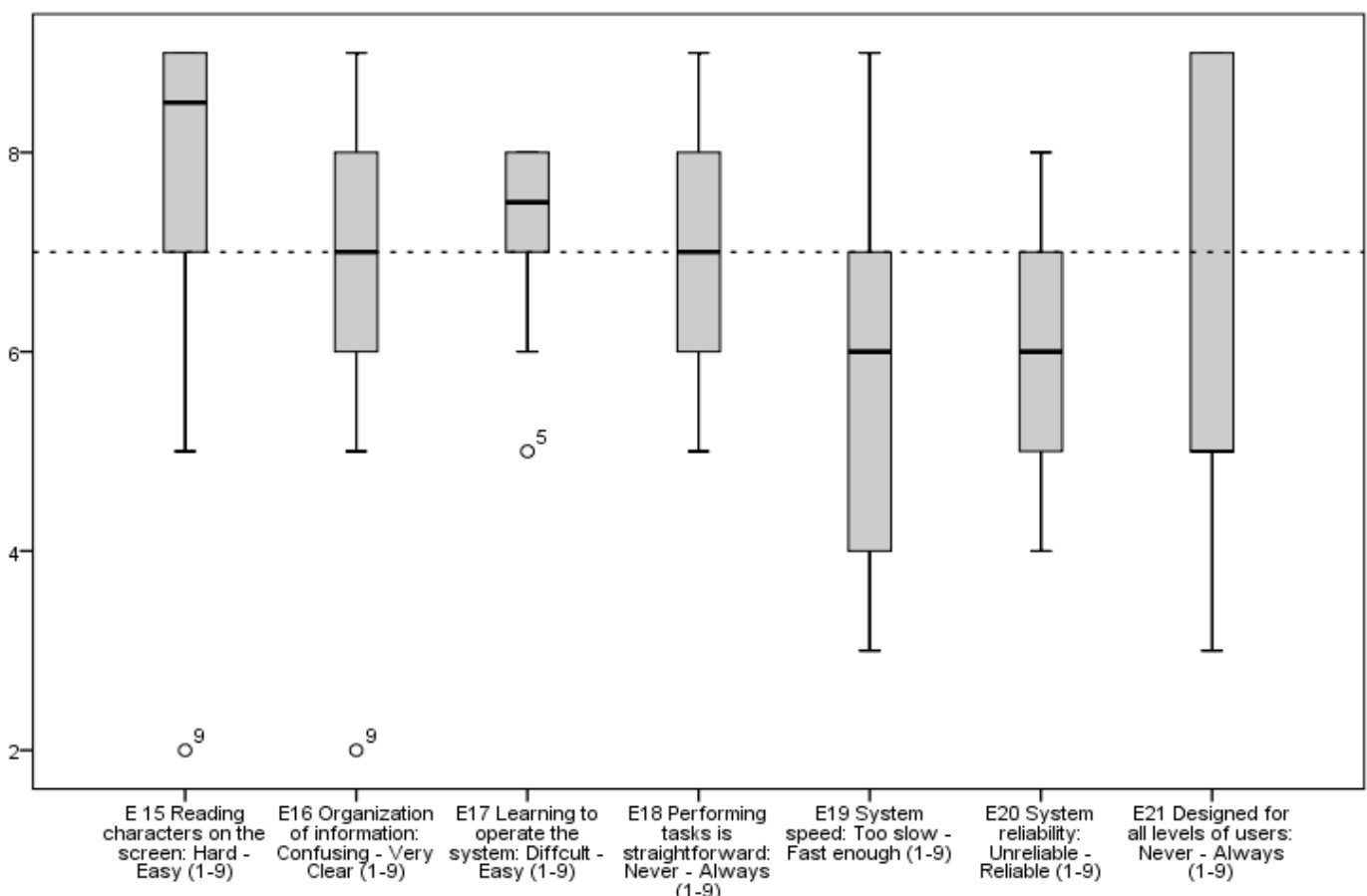

**Figure 12.** Boxplots representing the distribution of Examiner opinions on the screening platform (E15–E21). For each question, the black horizontal line inside the box represents the median score expressed, and the grey box represents the extension from the first to the third quartile. The dotted line represents the median score for the whole scale.

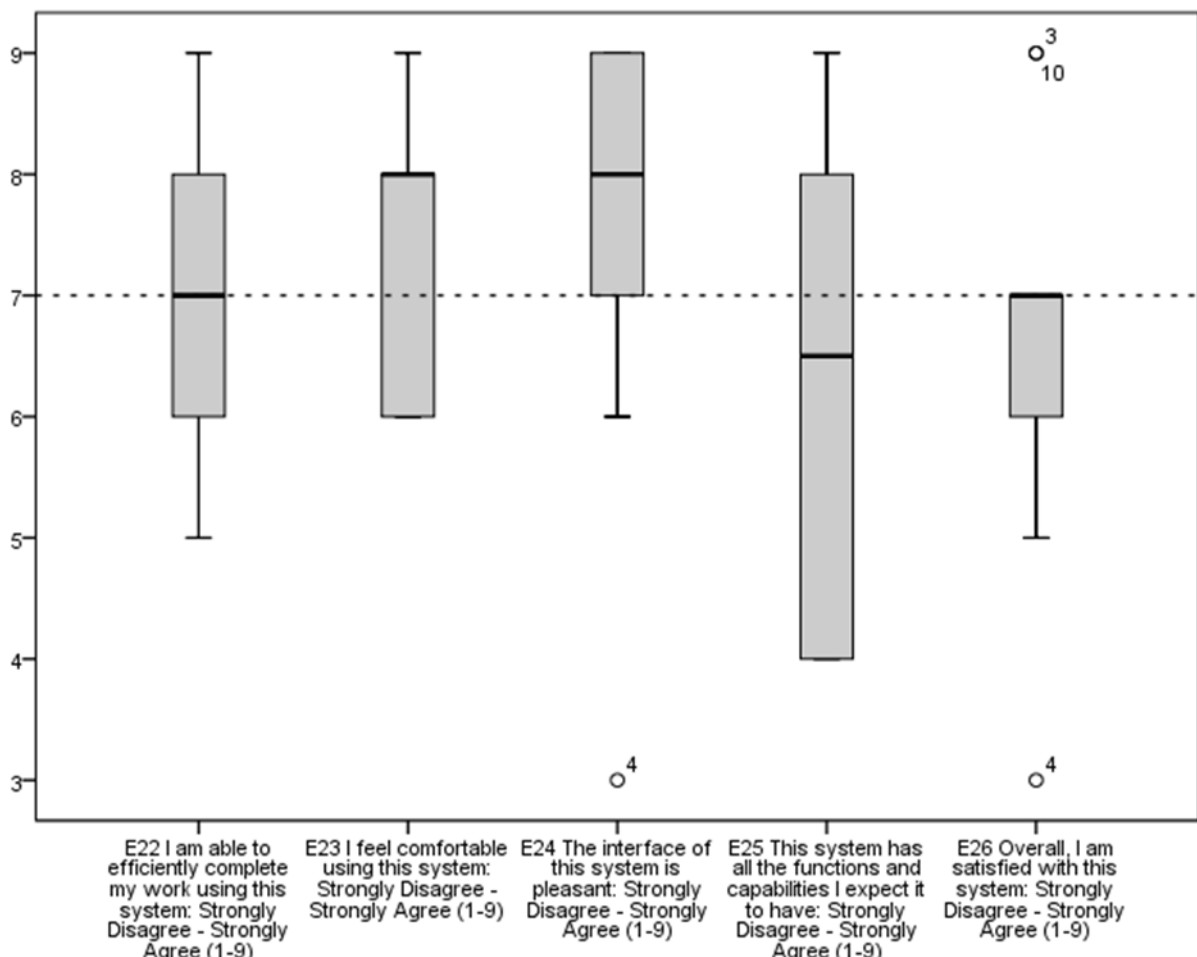

**Figure 13.** Boxplots representing the distribution of Examiner opinions on the screening platform (E22–E26). For each question, the black horizontal line inside the box represents the median score expressed, and the grey box represents the extension from the first to the third quartile. The dotted line represents the median score for the whole scale.

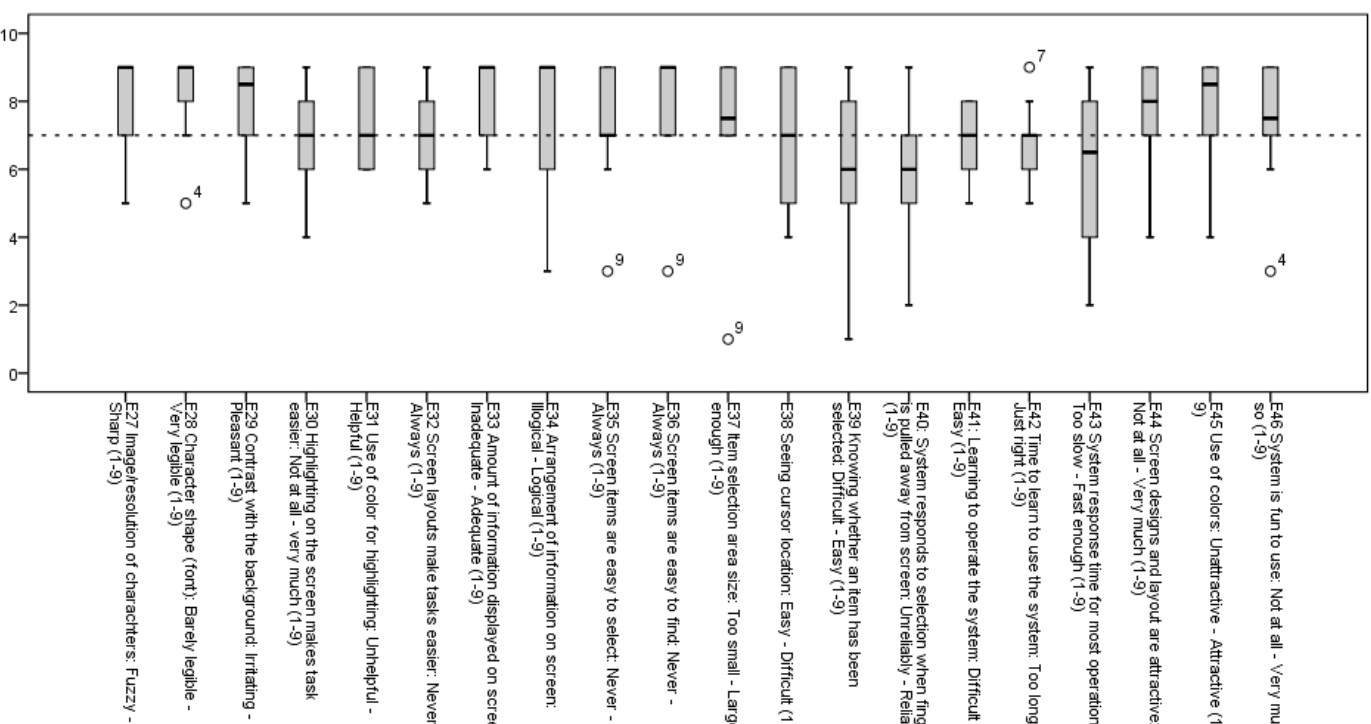

**Figure 14.** Boxplots representing the distribution of Examiners' opinions on the screening platform (E27–E46). For each question, the black horizontal line inside the box represents the median score expressed, and the grey box represents the extension from the first to the third quartile. The dotted line represents the median score for the whole scale.

## 4. Discussion

The current study provides evidence that it is (a) possible and (b) useful to follow the recommendations suggested by various policy oriented projects [18,22]. Indeed, this study not only confirms that children tested by means of computerized reading screenings are pleased with the medium (cf. [25]) but also adds direct evidence from a usability study.

### 4.1. Validity of the Screening Tasks

Similarly to previous studies (cf. [27,28]), performance in the screening tasks was found to be correlated with reading performance as measured by standardized reading tests [41,42]. In order to find out whether not only the word and nonword reading tests in Italian but also the L1 reading tests reflect bilingual children's general reading skills, the associations between these two measures of reading ability were assessed and found to be significant.

The results in most, although not all reading tasks of the MuLiMi screening battery were significantly associated with the performance in (subtests of) the standardized/traditional reading tests, both in the whole group and in the different subgroups. This finding confirms what is usually found in terms of parallel development of reading abilities in the L1 and the L2 and of transfer between the two languages [17–19].

Nonetheless, it should also be noted that the bilingual children in our sample were all schooled in both languages, either in mainstream school (English–Italian children) or in separate weekdays and weekend school (Mandarin–Italian children), so it is not possible to generalize this finding to the whole population of bilingual children. More precisely, the reading time scores measured in the self-paced syllable and sentence reading tasks along with performance in Word and Nonword Identification tasks of the Italian screening were significantly associated with the performance in the Italian standardized reading tests.

However, not only reading accuracy and reading speed in both languages (L1 and L2) but also other clinical markers proved to be useful for dyslexia risk identification, as suggested by previous studies [19,20,25–33,36–39]. Indeed, Phonological Blending and grammaticality judgement tasks were all significantly and positively associated with the performance in the Italian standardized reading tests. For the two groups of bilingual children, this general pattern was confirmed when comparing the screening task results to the risk levels derived from the standardized L1 reading test.

Since the results obtained in these screening tasks are also significantly associated with the level of DD-risk as revealed in the parental questionnaires (with the exception of the clitic judgment task, for which negative correlations emerged), it can be concluded that the tasks are suitable for the identification of DD risk in mono- and bilingual children. Turning to the teacher questionnaire, even if the only significant association emerging was with accuracy in the Italian Word Identification, this result confirms the validity of the screening with respect to restricted but crucial reading abilities.

Not only performance in the Italian screening tasks but also the performance in screening tasks in the children's L1 were significantly associated with standardized/traditional reading tests and risk scores. For the English screening tasks, similarly to the Italian screening tasks, reading time in the self-paced sentence reading task was significantly associated with word and nonword reading time in the English standardized reading test.

Furthermore, the associations between the standardized test scores and the English Orthographic Form Identification task as well as in the identification of correct tense marking suggest that also these screening tasks are suitable for the identification of DD-risk in English–Italian speaking children. This pattern was confirmed when comparing the children's screening task performance to the risk level derived from Italian and English standardized reading tests.

The pronunciation and deletion tasks were shown to be contributing to the detection of DD-risk, even though to a much smaller extent. For the Mandarin screening tasks instead, performance in the Onset and Rhyme Detection tasks as well as the Left–Right Inversion and Radical Position judgement tasks were associated with the number of correctly read Mandarin characters.

Mandarin RAN (digits) was also found to be suitable for DD-risk identification in Mandarin–Italian-speaking children. This finding is in line with studies suggesting that phonemic awareness plays a central role in the development of reading abilities in both alphabetic and logographic orthographic systems, even if the relative weight may vary slightly [21,22,29], and that naming deficits (RAN) as well as visual-perceptual skills play an additional role especially in transparent alphabetic orthographies, such as Italian or in logographic systems, such as Chinese.

Altogether, these results show that it is possible to automatically assess reading and reading-related skills in the child's L1 without any requirement for the Examiner to speak the language of assessment. It is worth noting that this information cannot directly lead to the diagnosis of a specific learning disorder, but it highlights the presence of an increased risk of DD and the need of in-depth investigation for diagnostic purposes.

The associations between performance in standardized tests and scores obtained in some of the screening tasks have several practical implications. First, these associations show that, in general, it is possible to assess general reading abilities of mono- and bilingual children using computerized screening tasks implemented on the screening platform MuLiMi.

In order to better understand the use and usefulness of the application of such automated screenings, the requirements for measuring reading time and accuracy in task performance need to be discussed separately. The measurement of reading time, necessary to evaluate a child's reading fluency, in clinical practice is usually achieved by manually measuring reading time using a stopwatch or similar and then calculating speed. The use of self-paced reading on the computer is a sufficiently proximal equivalent of this kind of measurement.

As for the measurement of accuracy, the usual practice for clinicians is to record and possibly classify each error produced by the child during reading and then assigning a score that may differ according to the type of error (e.g., DDE-2 scoring rules). Automatized tasks recording errors are not only less time consuming; however, they are often the only way to assess a child's language and/or reading skills in L1 without speaking that language.

### 4.2. Usability of the Web-Platform and Screenings

The usability study highlighted very positive results, confirming that the screening platform and tasks are enjoyable, easy to use and useful. While previous studies had already highlighted the potential of (a) computerized screenings for the identification of risk for dyslexia [27,28] and (b) remote testing solutions [14,26], the application of the latter is often restricted in terms of devices, operating systems and testing languages. As shown by the results of the present study, the MuLiMi platform offers a very flexible solution that can easily be adapted to different needs, situations and devices. Nonetheless, the usability study also highlighted some issues that should be addressed in future work.

The first issue appears to be reaction speed, both for Examinees (P5) and Examiners (E19). Regarding the Examinee interface, preliminary analysis suggests that network latency could be the main cause of the slow reaction time of the platform The Examinee's interface appears to be reactive enough on cabled networks when tested, and thus it is hypothesized that Examinees could use the platform from a slower network more often than we expected.

As to the slow loading of data to display into the tables in the Examinee interface, the cause of the delay is probably the method used to load the data: those are loaded in an eager way (all the data of the table are loaded at once and processed for rendering as well). A lazy loading method should drastically reduce the table's loading times.

Some of the Examiners mentioned the lack of a feature that would allow them to select the starting task of a test. The need behind this request is to be able to resume a forcefully interrupted test session, a problem that manifested frequently especially in remote sessions and should be solved by finding a way for detecting a non-completed test session and allow the Examiners to resume it.

Furthermore, the platform is perceived as slightly rigid by the Examiners. This was expected, and the reason likely lies in the platform design itself: the freedom of the Examiner has been restricted to increase the safety of operations. A post-hoc statistical analysis highlighted a significant ($n = 10$, $r = 0.837$, $p = 0.003$) correlation between the two reported complaints, suggesting that the perceived rigidity is related to missing expected features. A resume feature could thus reduce the perceived rigidity.

Finally, the Examiners highlighted that the platform may not be suited for some levels of users. Some of the interviewed clarified that some of the tests appeared too hard for the target Examinees. This issue should be solved by the envisaged process of task and item selection, that will be performed based on the results of preliminary validation.

### 4.3. Limitations and Future Perspectives

Despite the potential of the screening platform revealed in the previous sections, there are limitations regarding the screening platform and the study. The remote testing feature revealed a particular limitation: the STUN protocol has issues with some network configurations, especially when the Examinee tries to connect by an institutional network (for example the one of a school), which are usually protected by strong firewalls. WebRTC offers TURN as a solution for this issue. However, the TURN protocol requires one or more support server processes, and the cost to deploy those processes is rather high compared to expected benefits.

Furthermore, samples of tested children were small, which only partly allows for a generalization of the results. Moreover, the degree to which the clinical value of the screening platform can be assessed is very limited due to the low number of children with known specific reading disorders or at-risk for the latter in the present sample. Conclusions

on the potential in DD-risk evaluation were merely based on the risk that was inferred from performance on standardized tests, while no in-depth clinical investigation was conducted.

Therefore, the present study cannot be considered as a full validation study, but rather as a preliminary validation, which provided generally positive results for all aspects taken into consideration. Further studies need to be conducted in order to find out whether a combination of methods (indirect like parental and teacher questionnaires and direct testing of language performance in all languages spoken) can lead to risk detection better than single methods.

Based on these preliminary results, systematic item selection procedures in the most sensitive tasks should be conducted in order to further shorten the screening so as to improve its applicability further. This, combined with improved system speed/response, will facilitate future studies and the implementation of such screenings in classroom and clinical settings.

## 5. Conclusions

The current study highlighted that the MuLiMi platform provides a solution for various languages or combinations of languages (which can be expanded in the future), is usable and that the screening can also be carried out remotely. In particular, despite limitations in the system speed and in the ease of familiarization with the system for novel users, the functions and the format of the MuLiMi screening platform were well received both by Examiners and Examinees.

Moreover, the present results confirmed that the tasks implemented on the platform have good preliminary validity and allow discrimination of different degrees of risk in the children tested. Providing an easy-to-use and pleasant screening system and a series of discriminative tasks, it can be concluded that such computerized screenings, also carried out remotely, can be used for efficient and uncomplicated identification of the risk of reading difficulties in bilingual children, which in turn will contribute to increased opportunities for early intervention.

**Author Contributions:** All co-authors contributed to the manuscript. Conceptualization, M.L.L. and M.E.; Methodology, M.L.L.; Online screening platform implementation, M.S. and F.V.; Formal Analysis, M.L.L.; Investigation, M.E. and M.L.L.; Resources, M.E., M.S. and M.L.L.; Data Curation, G.M., M.E. and M.L.L.; Writing—Original Draft Preparation, M.E. and M.S.; Writing—Review and Editing, M.E., M.S., G.M., M.T.G., F.V., F.G. and M.L.L.; Visualization, M.S., M.E. and M.L.L.; Supervision, M.L.L. and F.G.; Project Administration, M.L.L. and M.T.G.; Funding Acquisition, M.L.L. and M.T.G. All authors have read and agreed to the published version of the manuscript.

**Funding:** This project received funding from the European Union's Horizon 2020 program for research and innovation under the Marie Skłodowska Curie Grant Agreement No. 765556 and by the Italian Ministry of Health, Grant RC2021 to Maria Luisa Lorusso.

**Institutional Review Board Statement:** The study (MultiMind project, id. number 438) was conducted according to the guidelines of the Declaration of Helsinki and approved by the Ethics Committee of the Scientific Institute Eugenio Medea, scientific section of the association "La Nostra Famiglia" Prot. N. 43/19, 17 June 2019.

**Informed Consent Statement:** All the participating children's parents or legal tutors gave their informed consent before they participated in the study. The study was conducted in accordance with the Declaration of Helsinki, and the protocol was approved by the Ethics Committee of the Scientific Institute Eugenio Medea, scientific section of the association "La Nostra Famiglia" Prot. N. 43/19, 17 June 2019.

**Data Availability Statement:** The database with all the usability data have been uploaded on Zenodo and will be made publicly available on publication. Database containing usability data of Participants:10.5281/zenodo.5661109, database containing usability data of Examiners: 10.5281/zenodo.5660197.

**Acknowledgments:** We would like to thank all the families and teachers who participated in the study, as well as the speech and language therapists and psychologists who took part in the usability

evaluation. We especially thank the directors and teachers from Acorn House, Rome, and Kindergarten, Florence. Furthermore, we would like to thank Shari Gandelli and Carolina Zanaletti who supported the recruitment, data collection and scoring process; Andrea Bigagli and Shenai Hu who created the Italian and Mandarin stimuli for previous projects; Theo Marinis and Sheila Keeshan for their help in the English item selection, creation and recording; Manvi Aggarwal, Riley Towers, Mohsen Pourshirazi, Mahsa Sedghi, Chen Yupeng, Zhou Xin, Zhao Yun, Mohammad Rahbari Solout and Lukasz Moskwa for their support in the development of the platform.

**Conflicts of Interest:** The authors declare no conflict of interest. The funding parties had no role in the design of the study; in the collection, analyses, or interpretation of data; in the writing of the manuscript; or in the decision to publish the results.

## Appendix A

**Table A1.** Items contained in the Online Survey on Usability of MuLiMi for Examinees/Participants, adapted from [51–53].

| Question Code | Question | Response Scale (5-Point-Scale) |
|---|---|---|
| P1 [1] | I think that I would need the support of a technical personto be able to use this system | Strongly agree–Strongly disagree |
| P2 | Overall reaction to the Software | Terrible–Wonderful |
| P3 | Overall reaction to the Software | Dull–Stimulating |
| P4 | Learning to operate the system | Difficult–Easy |
| P5 | System speed | Too slow–Fast enough |
| P6 | It was simple to use this system | Strongly Disagree–Strongly Agree |
| P7 | I feel comfortable using this system | Strongly Disagree–Strongly Agree |
| P8 | Characters on the touch screen | Hard to read–Easy to read |
| P9 | Screen elements are easy to select | Never–Always |
| P10 | See the cursor (arrow) position | Difficult–Easy |
| P11 | Feedback on whether an element is selected/clicked | Difficult–Easy |
| P12 | Are the screen graphics pleasant? | Not at all–Very much |

[1] In order to facilitate the interpretation of the responses, the response options of this question were inverted so that a higher score is always representative of a positive user experience.

## Appendix B

**Table A2.** Items contained in the Online Survey on Usability of MuLiMi for Examiners, adapted from [51–53].

| Question Code | Question | Response Scale | Scale |
|---|---|---|---|
| E1 | How many times have you administered a screeningusing the MuLiMi screening platform? | once<br>once to 5 times<br>more than 5 times | N/A |
| E2 [1] | I found the system unnecessarily complex | Strongly Disagree–Strongly Agree | 1–5 |
| E3 | I think the system was easy to use | Strongly Disagree–Strongly Agree | 1–5 |
| E4 [1] | I think that I would need the support of a technical personto be able to use this system | Strongly Disagree–Strongly Agree | 1–5 |
| E5 | I found the various functions of this system to be well integrated | Strongly Disagree–Strongly Agree | 1–5 |
| E6 | I would imagine that most people would learn to use this system very quickly | Strongly Disagree–Strongly Agree | 1–5 |

**Table A2.** *Cont.*

| Question Code | Question | Response Scale | Scale |
| --- | --- | --- | --- |
| E7 [1] | I found the system very cumbersome to use | Strongly Disagree–Strongly Agree | 1–5 |
| E8 | I felt very confident using the system | Strongly Disagree–Strongly Agree | 1–5 |
| E9 [1] | I needed to learn a lot of things before I could get goingwith this system | Strongly Disagree–Strongly Agree | 1–5 |
| E10 | Overall reaction to the Software | Terrible–Wonderful | 1–9 |
| E11 | Overall reaction to the Software | Difficult–Easy | 1–9 |
| E12 | Overall reaction to the Software | Frustrating–Satisfying | 1–9 |
| E13 | Overall reaction to the Software | Dull–Stimulating | 1–9 |
| E14 | Overall reaction to the Software | Rigid–Flexible | 1–9 |
| E15 | Reading characters on the screen | Hard–Easy | 1–9 |
| E16 | Organization of information | Confusing–Very Clear | 1–9 |
| E17 | Learning to operate the system | Difficult–Easy | 1–9 |
| E18 | Performing tasks is straightforward | Never–Always | 1–9 |
| E19 | System speed | Too slow–Fast enough | 1–9 |
| E20 | System reliability | Unreliable–Reliable | 1–9 |
| E21 | Designed for all levels of users | Never–Always | 1–9 |
| E22 | I am able to efficiently complete my work using this system | Strongly Disagree–Strongly Agree | 1–9 |
| E23 | I feel comfortable using this system | Strongly Disagree–Strongly Agree | 1–9 |
| E24 | The interface of this system is pleasant | Strongly Disagree–Strongly Agree | 1–9 |
| E25 | This system has all the functions and capabilities I expect it to have | Strongly Disagree–Strongly Agree | 1–9 |
| E26 | Overall, I am satisfied with this system | Strongly Disagree–Strongly Agree | 1–9 |
| E27 | Image/resolution of characters | Fuzzy–Sharp | 1–9 |
| E28 | Character shape (font) | Barely legible–Very legible | 1–9 |
| E29 | Contrast with the background | Irritating–Pleasant | 1–9 |
| E30 | Highlighting on the screen makes task easier | Not at all–Very much | 1–9 |
| E31 | Use of color for highlighting | Unhelpful–Helpful | 1–9 |
| E32 | Screen layouts make tasks easier | Never–Always | 1–9 |
| E33 | Amount of information displayed on screen | Inadequate–Adequate | 1–9 |
| E34 | Arrangement of information on screen | Illogical–Logical | 1–9 |
| E35 | Screen items are easy to select | Never–Always | 1–9 |
| E36 | Screen items are easy to find | Never–Always | 1–9 |
| E37 | Item selection area size | Too small–Large enough | 1–9 |
| E38 [1] | Seeing cursor location | Easy–Difficult | 1–9 |
| E39 | Knowing whether an item has been selected | Difficult–Easy | 1–9 |
| E40 | System responds to selection when finger is pulled away from screen | Unreliably–Reliably | 1–9 |
| E41 | Learning to operate the system | Difficult–Easy | 1–9 |
| E42 | Time to learn to use the system | Too long–Just right | 1–9 |
| E43 | System response time for most operations | Too slow–Fast enough | 1–9 |

**Table A2.** *Cont.*

| Question Code | Question | Response Scale | Scale |
|:---:|:---:|:---:|:---:|
| E44 | Screen designs and layout are attractive | Not at all–Very much | 1–9 |
| E45 | Use of colors | Unattractive–Attractive | 1–9 |
| E46 | System is fun to use | Not at all–Very much so | 1–9 |

[1] In order to facilitate the interpretation of the responses, the response options of this question were inverted so that a higher score is always representative of a positive user experience.

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
