# Peer review of "Remote Dyslexia Screening for Bilingual Children"

_mti, doi:10.3390/mti6010007_

Round 1

Reviewer 1 Report

Your introduction must present the theme under investigation, the objectives and the research questions. I felt that your current introduction is too structured. I do not understand why to organise the whole subsection consisting of one paragraph only:

1.1. Bi- and Multilingualism 33
Multilingual persons are confronted with more than one language in their everyday 34 life [1]. While the language used in the family context often is the first language acquired, 35 it is referred to as L1, while the language used outside home is referred to as societal lan-36 guage or L2 since the languages are usually acquired sequentially. Besides sequential mul-37 tilingual language acquisition, the different languages can also be acquired simultane-38 ously. Language competence can vary between the languages spoken. Generally speak-39 ing, multilingual language acquisition does follow the same stages as monolingual lan-40 guage acquisition, but can vary in terms of timing [2].

At the end of your introduction your reader would expect to see the article outline. I would give a summary of it.

Under 4. Discussion pleease discuss the most salient outcomes in light of the theme and literature.

Please be very precise when writing. Please provide references to support your statements. For example, here:

5. Conclusions 877
While previous studies had already highlighted the potential of (a) computerized 878 screenings for the identification of risk for dyslexia and (b) remote testing solutions, the 879 application of the latter is often restricted in terms of devices, operating systems and test-880 ing languages.

Your reader wishes to see references to "previous studies had already highlighted the potential of..."

Reviewer 2 Report

This study looked at the remote screening of dyslexia among bilingual children residing in Italy. It is an interesting study and clearly fills a gap in the literature around screening of dyslexia among children speaking more than one language. In general, the manuscript is well written and data clearly presented. However, I do think some changes are needed. In the introduction for example, there is very little information on bilingualism:

Lines 35-37, the authors suggest that languages are usually acquired sequentially. However, this is a very strong statement and not necessarily true. Many bilinguals acquire their languages simultaneously, for example, in countries where multiple languages are spoken. I would rephrase this sentence. Also, this section on bi- and multilingualism is very short and lacks references. I would suggest adding some more information on bilingualism and adding some references.

Page 1, line 43, the authors claim that learning to read and write starts at five to six years of age. However, children develop emergent literacy skills much earlier, and these contribute to learning to read and write around the age of 5 or 6. I would stress that most children learn to 'formally' read and write after having received instruction in school as children arrive at school with different reading and writing abilities. 

I think it is a big jump to automatically label all children who have specific reading disorders as having developmental dyslexia as there are many reasons why children struggle with reading and/or writing.

Page 2, line 70, I would explain what 'gamified' testing is.

Line 150: why and how were these children chosen?

Line 204: are the instructions written or spoken?

Line 235: the Italian screening tasks, where these standardised tasks? Same for tasks in other languages? What were these based upon, is there information available on reliability and validity?

Line 471: why use so many different versions of parental questionnaires and not let all parents fill in questionnaire in same format?

Instead of the boxplots, I would just report the results or provide them in a chart as the boxplots are not always easy to read.

I would shorten the section on satisfaction with the user experiences and remove the boxplots.

The discussion should be an interpretation of the results, currently it reads more like a repetition of the results instead of how this study contributes to the literature. Why is this study important? How does it fit within the literature.
